# Analysis of human total antibody repertoires in TIF1γ autoantibody positive dermatomyositis

Spyridon Megremis [1,9], Thomas D. J. Walker [2,9], Xiaotong He [2], James O'Sullivan[1], William E. R. Ollier[3,4], Hector Chinoy [5,6], Neil Pendleton [7], Antony Payton[8], Lynne Hampson[2], Ian Hampson[2,10] & Janine A. Lamb [3,10 ✉]

We investigate the accumulated microbial and autoantigen antibody repertoire in adult-onset dermatomyositis patients sero-positive for TIF1γ (TRIM33) autoantibodies. We use an untargeted high-throughput approach which combines immunoglobulin disease-specific epitope-enrichment and identification of microbial and human antigens. We observe antibodies recognizing a wider repertoire of microbial antigens in dermatomyositis. Antibodies recognizing viruses and Poxviridae family species are significantly enriched. The identified autoantibodies recognise a large portion of the human proteome, including interferon regulated proteins; these proteins cluster in specific biological processes. In addition to TRIM33, we identify autoantibodies against eleven further TRIM proteins, including TRIM21. Some of these TRIM proteins share epitope homology with specific viral species including poxviruses. Our data suggest antibody accumulation in dermatomyositis against an expanded diversity of microbial and human proteins and evidence of non-random targeting of specific signalling pathways. Our findings indicate that molecular mimicry and epitope spreading events may play a role in dermatomyositis pathogenesis.

[1] Division of Evolution and Genomic Sciences, University of Manchester, Manchester, UK. [2] Division of Cancer Sciences, University of Manchester, Manchester, UK. [3] Division of Population Health, Health Services Research & Primary Care, University of Manchester, Manchester, UK. [4] Centre for Bioscience, Faculty of Science and Engineering, Manchester Metropolitan University, Manchester, UK. [5] National Institute for Health Research Manchester Biomedical Research Centre, Manchester University NHS Foundation Trust, University of Manchester, Manchester, UK. [6] Department of Rheumatology, Salford Royal NHS Foundation Trust, Manchester Academic Health Science Centre, Salford, UK. [7] Division of Neuroscience & Experimental Psychology, University of Manchester, Manchester, UK. [8] Division of Informatics, Imaging & Data Sciences, University of Manchester, Manchester, UK. [9] These authors contributed equally: Spyridon Megremis, Thomas D. J. Walker. [10] These authors jointly supervised this work: Ian Hampson, Janine A. Lamb. ✉email: Janine.Lamb@manchester.ac.uk

The idiopathic inflammatory myopathies (IIM) are a heterogeneous spectrum of rare autoimmune musculoskeletal diseases characterised clinically by muscle weakness and systemic organ involvement. IIM are thought to result from immune activation following environmental exposures in genetically susceptible individuals. Viral and bacterial infections have been reported in IIM[1], but their role in disease pathology is unclear.

Autoantibodies are a key feature of IIM, in common with other autoimmune rheumatic diseases such as rheumatoid arthritis, systemic lupus erythematosus and systemic sclerosis. Myositis-specific autoantibodies, present in 60–70% of individuals with IIM, are directed against cytoplasmic or nuclear components involved in key intracellular processes, including protein synthesis and chromatin re-modelling[2]. Myositis-specific autoantibodies are often associated with particular clinical features. Individuals with autoantibodies targeting the cytoplasmic nucleic acid sensor MDA5 (also called interferon induced with helicase C domain 1, IFIH1) can present with rapidly progressive interstitial lung disease associated with high mortality[2,3]. Several members of the tripartite motif (TRIM) protein family are autoantibody targets in IIM, and there is a strong temporal association between adult–onset dermatomyositis and malignancy onset in individuals with antibodies to transcription intermediary factor 1γ (TIF1γ, TRIM33)[4]. Many TRIM proteins are important immune regulators[5,6], and dysregulation of TRIM proteins leading to reduced ability to restrict viral infection has been reported in autoimmune diseases including systemic lupus erythematosus and inflammatory bowel disease[7,8].

The inducible type I interferon (T1-IFN) cytokine system, part of the innate immune response, also plays a role in autoimmune rheumatic diseases including IIM[9]. The T1-IFN antiviral response is initiated when pathogen-associated molecular patterns are recognised by host pattern recognition receptors and cytosolic receptors for viral nucleic acid. This broad viral recognition process triggers downstream signalling pathways which lead to interferon transcription, protein production and expression of interferon-induced genes; this further enhances the antiviral machinery[9]. Such a response is critical for host protection against pathogen expansion and reduces infection during the window of time needed to mount an effective specific (adaptive) immune response.

Improved understanding of IIM pathogenesis is required to improve both patient stratification and disease management. In adult–onset dermatomyositis, we propose the following potential molecular mechanism of disease pathogenesis: anti-TIF1 autoantibodies reduce the ability to restrict viral infection, which leads to either increased susceptibility to a wider diversity of viral pathogens or increased exposure to specific anti-TIF1-related viruses. To test this hypothesis, we applied a novel high-resolution and high-throughput comparative screening pipeline (serum antibody repertoire analysis, SARA) [paper in preparation] to anti-TIF1 autoantibody-positive dermatomyositis patients (DM) and matched healthy control (HC) plasma. Using this approach, high-throughput antigen epitope-sequencing was integrated with bioinformatic modules to de-convolute accumulated immunogenic responses against the total microbial "exposome" (including viruses, bacteria, archaea and fungi) and human proteins. We report the identification of disease-associated microbial and human protein epitopes which have clinical and aetiological relevance to anti-TIF1 autoantibody-positive dermatomyositis.

## Results

To describe the accumulated antibodies present in dermatomyositis we use the SARA pipeline which integrates an *Escherichia coli* FliTrx™ random 12 amino acid (AA) peptide display system with epitope signature enrichment through competitive bio-panning and high-throughput DNA sequencing (Fig. 1). Competitive bio-panning was applied to pooled total immunoglobulin fractions (IgA, IgG and IgM) purified from the plasma of 20 anti-TIF1 positive adult–onset DM and 20 HC (Supplementary Table 1). Four sample pools were generated and paired: the first pair contained 10 pooled samples (P10) of DM (DM P10) used for competitive biopanning against 10 pooled samples of HC (HC P10) (Supplementary Table 1). The second pair included 20 pooled samples (P20) of DM (DM P20) vs. 20 pooled samples of HC (HC P20) (Supplementary Table 1). We used two sample-pools of increasing sample sizes so that we could evaluate the heterogeneity of our observations within the DM and HC groups. Figure 1 and Supplementary Figure 1 detail achieved metrics while defining the microbial and autoantibody immunogenic repertoires in DM and HC. We retrieved ≈36 million (DM) and ≈24 million (HC) next-generation sequencing (NGS) reads which represent 8.7 million (DM) and 4.7 million (HC) expressed epitopes (Supplementary Data 1). Cohorts presented highly enriched epitope sequences that are unique to DM (15,522) and HC (4817), respectively. Our epitope cohorts retrieved 6.75 million (DM) and 2.25 million (HC) microbial (Supplementary Data 2) or human protein annotations (Supplementary Data 3) which mapped to 9111 (DM) and 4994 (HC) *Distinct* species. Totally, 6202 and 2085 *Unique* species were identified in DM and HC, respectively.

**Wider repertoire of antibodies recognising microbial antigens in dermatomyositis**. In DM, antibodies against linear microbial epitopes were identified for a total of 1560 microbial species (Supplementary Fig. 2a) compared to 1498 in HC (Supplementary Fig. 2b). In both groups, the highest richness (number of different species) was observed for bacteria, followed by viruses, fungi and archaea (Supplementary Fig. 2a, b). We defined *distinct* epitopes as any discrete epitope sequence retrieved from the DM and HC groups. *Unique* distinct epitopes represent any discrete epitope sequences that were observed exclusively in either DM or HC. Common distinct epitopes were discrete epitope sequences observed in both DM and HC by the sequence at a minimum of fivefold the NGS enrichment frequency in one group vs. the other. Unique distinct epitopes represented the vast majority of sequences observed (Fig. 1). Based on the distribution of the corresponding microbial species in the pools of 10 and 20 the majority of distinct species are shared between the P20 and P10 samples, whereas the majority of unique species are observed either in P20 or in P10, in both DM and HC (Supplementary Fig. 2c, d, respectively).

The increase in the number of plasma samples within the pool used for cross-panning had a differential effect between the two groups: in DM, an increase in the number of microbial species was observed from 728 (DM P10) to 832 (DM P20), whereas in the HC group a decrease was observed from 780 (HC P10) to 718 (HC P20). The number of epitopes per microbial species in both DM and HC significantly increased with increasing pooling size (Fig. 2a). The number of NGS reads per microbial species significantly increased in DM P20 (95% CI: 3.97–4.13) compared to DM P10 (95% CI: 3.90–3.98), whereas it did not change in the HC P20 (95% CI: 4.09–4.26) compared to the P10 (95% CI: 4.34–4.43) (Fig. 2b).

The number of distinct epitopes per microbe (epitopes present in both groups at minimum fivefold enrichment) significantly increased in the P20s compared to P10s in both groups (Fig. 2c); $n = 432$, (95% CI: 0.86–0.94) to $n = 455$ (95% CI: 2.02–2.10) in DM and $n = 432$ (95% CI: 1.29–1.37) to $n = 448$ (95% CI:

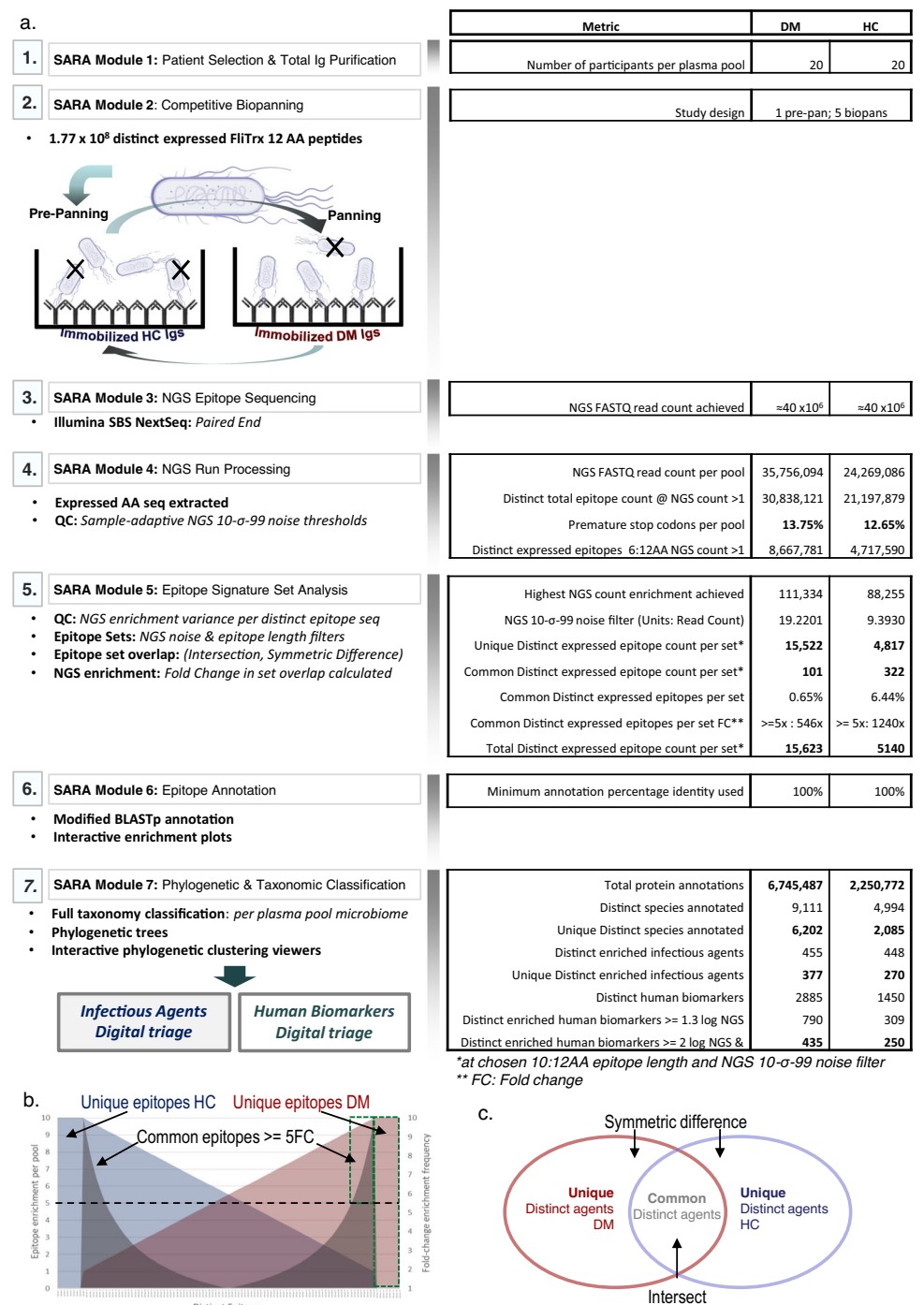

| Metric | DM | HC |
|---|---|---|
| Number of participants per plasma pool | 20 | 20 |
| Study design | 1 pre-pan; 5 biopans | |
| NGS FASTQ read count achieved | ≈40 x10⁶ | ≈40 x10⁶ |
| NGS FASTQ read count per pool | 35,756,094 | 24,269,086 |
| Distinct total epitope count @ NGS count >1 | 30,838,121 | 21,197,879 |
| Premature stop codons per pool | **13.75%** | 12.65% |
| Distinct expressed epitopes  6:12AA NGS count >1 | 8,667,781 | 4,717,590 |
| Highest NGS count enrichment achieved | 111,334 | 88,255 |
| NGS 10-σ-99 noise filter (Units: Read Count) | 19.2201 | 9.3930 |
| Unique Distinct expressed epitope count per set* | **15,522** | **4,817** |
| Common Distinct expressed epitope count per set* | **101** | **322** |
| Common Distinct expressed epitopes per set | 0.65% | 6.44% |
| Common Distinct expressed epitopes per set FC** | >=5x : 546x | >= 5x: 1240x |
| Total Distinct expressed epitope count per set* | **15,623** | **5140** |
| Minimum annotation percentage identity used | 100% | 100% |
| Total protein annotations | **6,745,487** | **2,250,772** |
| Distinct species annotated | 9,111 | 4,994 |
| Unique Distinct species annotated | **6,202** | **2,085** |
| Distinct enriched infectious agents | 455 | 448 |
| Unique Distinct enriched infectious agents | **377** | **270** |
| Distinct human biomarkers | 2885 | 1450 |
| Distinct enriched human biomarkers >= 1.3 log NGS | 790 | 309 |
| Distinct enriched human biomarkers >= 2 log NGS & | **435** | **250** |

*at chosen 10:12AA epitope length and NGS 10-σ-99 noise filter
** FC: Fold change

**Fig. 1 The serum antibody repertoire analysis (SARA) pipeline employed for plasma samples of dermatomyositis patients. a** SARA pipeline modules and metrics from this investigation. Module 1: Patient Selection and Total Ig Purification. Module 2: Competitive Biopanning. Pre-planned clones which bind immobilised healthy control (HC) Igs were discarded. Unbound clones passed immediately across immobilised dermatomyositis (DM) pool Igs. Retained clones contain amino acid (AA) epitope sequences of interest. Module 3: Next-generation sequencing (NGS) Epitope Sequencing. Biopanned-enriched epitopes were DNA sequenced (custom Illumina NextSeq process). Module 4: NGS Run Processing. Expressed peptide DNA variance regions were extracted and NGS noise removed (10-σ-99 threshold). Module 5: Epitope Signature Set Analysis. Epitope sequence frequency per AA length was determined. Distinct epitope set analysis determined unique distinct epitope pools within DM and HC. Distinct epitopes common to both cohorts were further segregated by fivefold NGS enrichment thresholds. Module 6: Epitope Annotation. Our modified BLASTp process annotated epitope sequences to every known protein. Module 7: Phylogenetic and Taxonomic Classification. Annotated Organisms were anchored by taxonomic classifications to contextualise the plasma pool microbial profiles in DM and HC cohorts. **b** Example schematic of enriched Distinct epitopes (blue: HC pool; Red: DM pool, grey: fold-change). Unique Distinct sets populate from epitopes exclusively in one pool (flanking blue/red areas; no fold-change overlay). Common Distinct sets populate from an epitope common to both pools with ≥fivefold enrichment differential (black dotted line). Green dotted boxes thus represent the sum of Unique Distinct and Common Distinct epitope sets from our DM analysis (not to scale). **c** Venn diagram of Unique Distinct (symmetric difference) and Common Distinct (intersect) infectious agents.

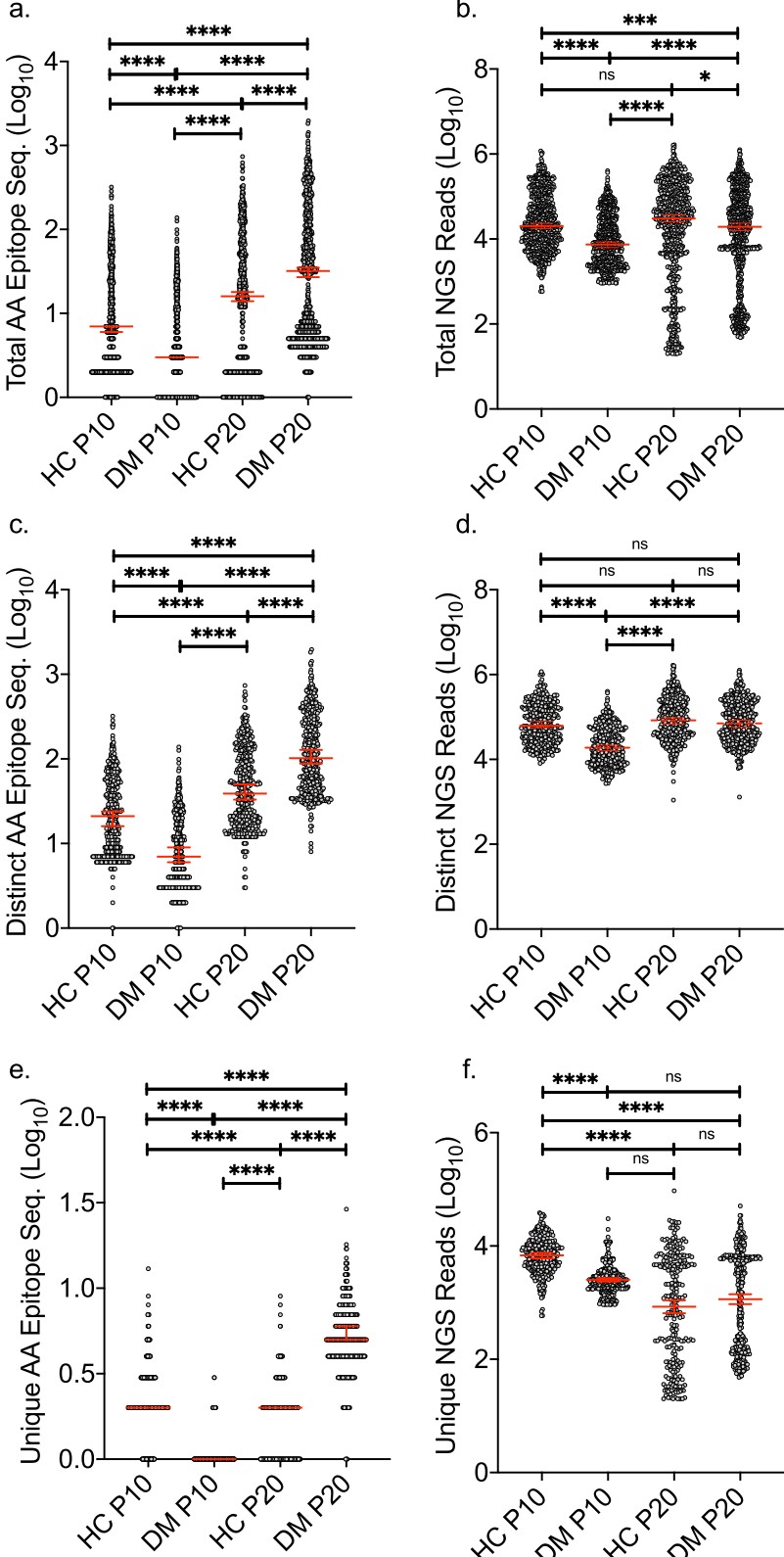

**Fig. 2 Microbial amino acid epitopes and their abundance in dermatomyositis and healthy controls.** The number of different epitopes per microbial species (left panel) and the number of next-generation sequencing (NGS) reads that mapped to these epitopes (right panel) are depicted; **a** Total number of microbial amino acid (AA) epitopes, **b** microbial NGS reads, **c** number of distinct AA epitopes, **d** distinct microbial NGS reads, **e** number of unique AA epitopes and **f** unique microbial NGS reads. All values were $\log_{10}$ transformed. Dunn's multiple comparisons test was used to test specific sample pairs. Differences were significant at the 0.05 level; *for $p < 0.05$, **for $p < 0.01$, ***for $p < 0.001$, etc. DM dermatomyositis, HC healthy controls, ns not-significant, P10 pool of 10, P20 pool of 20. The median with 95% CI is annotated.

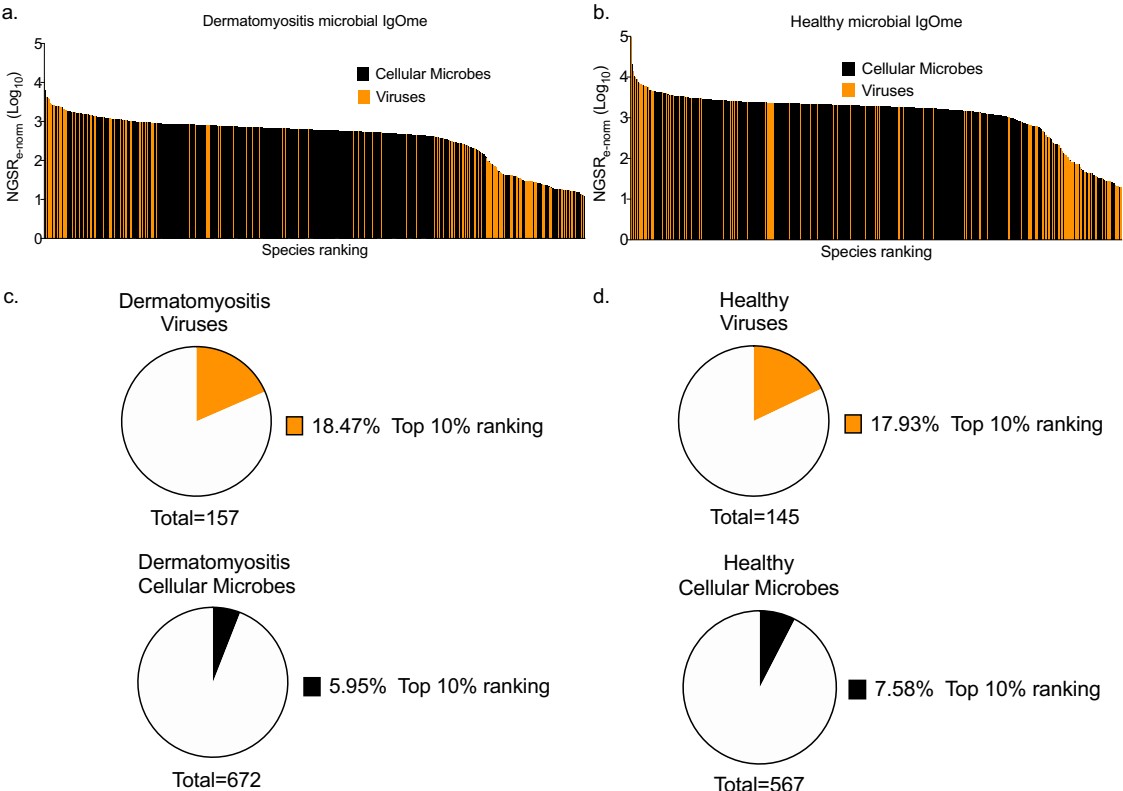

**Fig. 3 Ranking of identified microbial species based on the number of NGS reads and a number of epitopes.** Microbial species targeted by the identified antibodies were grouped in cellular microbes (Bacteria, Eukaryota and Archaea) and viruses and ranked based on the NGSR$_{e\text{-norm}}$. The distribution of members of the two groups can be seen in **a** dermatomyositis, and **b** healthy controls; viruses were superimposed over cellular microbes. The pie charts depict the proportion of each microbial group that is represented in the top 10% of ranked species in **c** dermatomyositis and **d** healthy controls. Orange: viruses, Black: cellular microbes.

1.60–1.68) in HC. The unique microbial epitopes which were identified only in one of the two groups significantly increased in DM P20 ($n = 377$, 95% CI: 0.69–0.77) compared to DM P10 ($n = 296$, 95% CI: 0.01–0.03) and decreased in HC P20 ($n = 270$, 95% CI: 0.18–0.23) compared to HC P10 ($n = 348$, 95% CI: 0.31–0.36) (Fig. 2e). Collectively, these data indicate an expansion of the DM-specific antibody repertoire against microbial antigens with increasing DM plasma sample size, whereas the opposite effect was observed in HC.

The number of enriched NGS reads per distinct microbial species significantly increased in DM P20 (95% CI: 4.83–4.92) compared to DM P10 (95% CI: 4.27–4.36), while no significant difference was observed in HC P20 (95% CI: 4.89–4.98) compared to HC P10 (95% CI: 4.80–4.89) (Fig. 2d). The number of sequencing reads per unique microbial species did not differ significantly between DM P20 and DM P10 (adjusted $p. > 0.999$), however, it significantly decreased in the HC P20 (95% CI: 2.81–3.04) compared to the HC P10 (95% CI: 3.78–3.85) (Fig. 2f). Overall, the differential effect of increased sample size on the observed antibody repertoire against microbial antigens between DM and HC suggests there is higher biological variability in DM than in HC.

DM P20 exhibited a significantly higher number of AA epitopes per microbial species compared to HC P20 (Fig. 2a) and a higher number of total identified microbial species (832 vs. 718). The same difference was observed in both distinct (Fig. 2c) and unique epitope sequences (Fig. 2e). Overall, we demonstrate that plasma from anti-TIF1 DM patients contains a higher number of microbial epitopes per species and against a wider microbial repertoire.

**Role of viruses in dermatomyositis.** We focused further analysis on the DM P20 and HC P20 subgroups. To stratify microbial species based on their potential significance in DM, we first normalised the number of NGS reads against the total number of epitopes per microbial species (NGSR$_{e\text{-norm}}$) (Supplementary Fig. 3a–d) and studied the relative ranking of viruses and cellular microbes (Fig. 3a, b). Secondly, we evaluated the mean NGSR$_{e\text{-norm}}$ at the viral family taxonomy level (Fig. 4d, e), and, thirdly, we recorded the total number of species contributing to each family (Fig. 4d, e) (Supplementary Table 2). Thus, we tested for the potential high taxonomy-level organisation of microbial antigens in DM. Ranking the microbial species based on decreasing NGSR$_{e\text{-norm}}$ we observed that viruses were over-represented in the top 10% of dominant microbial species relative to the total number of viral species present in both DM and HC (Fig. 3a, b). Specifically, 18.47% and 17.93% of viral species were present in the top 10% of microbes in DM P20 and HC P20, respectively (Fig. 3c, d). Cellular microbes were under-represented in the dominant species with 5.95% in DM P20 and 7.58% in HC P20 (Fig. 3c, d). Due to the competitive panning process, the over-representation of viral compared to cellular microbial species in the top-ranked microbes enriched in DM and HC suggests an important role of virus exposure in the environment–host immune cross-talk in DM.

**Poxviruses are tightly linked with dermatomyositis.** Viruses were grouped in high-order taxonomic groups based on their genome type (Fig. 4a, b). The HC viral IgOme profile contained an elevated proportion of total double-stranded DNA (dsDNA)

(NGSR$_{e-norm}$ 46.35% vs. 39.68%) and single-stranded RNA (ssRNA) (NGSR$_{e-norm}$ 43.70% vs. 35.85%) than DM (Fig. 4a, b). The DM viral IgOme contained an elevated proportion of ssDNA viruses (NGSR$_{e-norm}$ 15.95% vs. 3.10%) and RNA reverse-transcribing retroviruses (NGSR$_{e-norm}$ 7.16% vs. 3.19%) (Fig. 4a, b). However, the overall antibody virus IgOme compositional profiles between DM and HC were not significantly different (Wilcoxon matched-pairs $p$:0.843). We then compared the species-specific NGSR$_{e-norm}$ per viral category between DM and HC (Fig. 4c); The dsDNA (Mann Whitney two-tailed test $p < 0.000$) and ssRNA (Mann Whitney two-tailed test $p < 0.000$) NGSR$_{e-norm}$ were elevated in the HC plasma compared to DM, whereas no significant change was observed in ssDNA viruses (Unpaired $t$ test $p$:0.19) or RNA reverse-transcribing viruses (Mann Whitney two-tailed test $p$:0.273) (Fig. 4c).

Overall, antibodies against 47.9% (23 of 48) of viral families were represented in both groups. We compared the viral IgOme at the family level, and we recorded the mean NGSR$_{e-norm}$ with increasing enrichment for antibodies recognising each viral family (Fig. 4d, e). In DM, the 5 richest viral families were Coronaviridae (20 species), Geminiviridae (16 species), Herpesviridae (13 species), Orthomyxoviridae (9 species) and Poxviridae (8 species) (Fig. 4d) (Supplementary Table 2). Of these, Geminiviridae infects plants, whereas all other viruses can infect and cause disease in humans. In the HC, Picornaviridae (15 species) was the most enriched virus group followed by Caliciviridae (11 species), Orthomyxoviridae (9 species), Coronaviridae (9 species) and Retroviridae (8 species) (Fig. 4e) (Supplementary Table 2).

We evaluated the mean NGSR$_{e-norm}$ for each family. In DM, Nairoviridae ($n = 1$), Poxviridae ($n = 8$), Secoviridae (natural host: plants) ($n = 1$), Caliciviridae ($n = 4$) and Adenoviridae ($n = 1$) were the top-ranked families (Supplementary Table 3). Poxviridae was the one family that ranked highly regarding both the NGSR$_{e-norm}$ (Supplementary Table 3) and the family richness ($n = 8$, 95% CI: 2.98–3.61) (Supplementary Table 2). All eight pox viruses had high NGSR$_{e-norm}$ and were amongst the dominant DM viral species recognised (Fig. 4f). Specifically, *Variola* virus had the highest NGSR$_{e-norm}$ amongst all identified viral species (Fig. 4F). In the HC P20, Polyomaviridae ($n = 1$), Iflaviridae ($n = 1$), Podoviridae ($n = 1$), Tymoviridae ($n = 2$) and Myoviridae ($n = 3$) were the viral families with the highest mean NGSR$_{e-norm}$. Podoviridae and Myoviridae are prokaryotic viruses infecting bacteria, Tymoviridae infect plants and Iflaviridae infect insects (ViralZone root-ExPASy)[10]. We used the Z transformation of NGSR$_{e-norm}$ to define the precise location and rank of each viral family within the DM and HC distribution (Fig. 4d, e). In DM, multiple viral families (Nairoviridae, Poxviridae, Secoviridae, Alloherpesviridae, Caliciviridae and Adenoviridae) with high NGSR$_{e-norm}$ Z score occurred with at least one standard deviation above the group mean. In HC, only Polyomaviridae ($n = 1$) occurred with a high NGSR$_{e-norm}$ Z score. Supplementary Figure 4 depicts a cladogram of DM-specific viral species.

**Detection of autoantibodies against multiple TRIM proteins in dermatomyositis.** We queried the NGS reads against the human proteome to identify accumulated autoantibodies in the pooled plasma. In DM P20 we identified TRIM33 (Mean log$_{10}$:1.301), in accordance with the autoimmune profile of our selected plasma (Fig. 5a), which was not present in HC P20. We identified autoantibodies recognising 11 additional TRIM proteins in DM P20; TRIMs 21, 69, 47, 46, 27, 60, 10, 7, 77, 3 and TRIML2 (Fig. 5a). Of these, TRIM21 was also detected in HC but with significantly lower NGS reads, and TRIM25 (Mean log$_{10}$: 1.778) was observed only in HC P20 (Fig. 5a). These results confirm the

presence of TRIM33 autoantibodies in the selected twenty anti-TRIM33-positive DM patients and demonstrate the presence of autoantibodies against other members of the TRIM protein family exclusively in DM.

**An expanded dermatomyositis-specific and IFN-regulated human proteome.** From our autoantibody analysis, in DM P20 we identified antibodies against a total of 2885 human protein targets, of which 2537 were highly specific sequence annotation hits (Mean Sig. < 0.578). In HC P20 we detected 1450 human protein targets, 811 with high specificity (Mean Sig. < 0.55). Autoantibodies identified in both DM and HC constituted only 13% ($n = 498$) of total proteins and 8.4% ($n = 260$) of highly specific proteins suggesting a strong disease-associated proteome signature. The 2:1 (total proteins) and 3:1 (high specificity) ratios of identified autoantibodies in DM over HC suggest that an expanded subset of the human proteome is targeted by auto-antibodies in DM patients compared to HC.

Due to the role of TRIM proteins in IFN signalling, we asked whether the autoantibody protein-targets are regulated by interferons. In the DM dataset 1560 proteins were predicted to be regulated by IFNs compared to 518 in HC (Interferome v2.01)[11] suggesting a DM-specific enrichment of immunoglobulins against IFN-regulated human proteins. The vast majority of these proteins are regulated by interferons type I, type II or both (Supplementary Fig. 5a). We focused our search directly for IFN autoantibodies expressed in our samples. In DM, IFNGR1 autoantibodies were present, whereas they were absent in HC (Supplementary Fig. 5b). We expanded our search for auto-antibodies against known proteins that are highly ranked within the IFNG signalling pathway (gene rank within the SuperPath; GeneCardsSuite: PathCards)[12]. Autoantibodies against 26 IFNG-related proteins were observed in DM, whereas only 5 were found in HC (Supplementary Fig. 5b). The protein–protein interaction enrichment score in DM was $<1.0e-16$ vs. 0.123 in HC (STRING 11.0)[13] (Supplementary Fig. 5c, d). The ranking of TRIM proteins and INFG-related proteins based on sequence specificity and number of NGS reads is described in Supplementary Fig. 5e, f. Overall, these data suggest the accumulation of antibodies in DM against proteins that strongly contribute to IFNG signalling and the broader antiviral mechanism by IFN-regulated proteins.

To describe the biological functions of the identified proteins the Gene Ontology (GO) framework was used (Fig. 5b, c). In DM P20 eight GO biological processes were highly enriched (Fig. 5b). These processes were represented by an average of 25.6% of GO-specific proteins ($n = 996$, 95% CI: 21.2–29.7) (Supplementary Fig. 6a). In HC P20, ten GO processes were enriched with a lower average coverage of 14.5% ($n = 592$, range: 12.5–15.7), as fewer proteins mapped to the same GO unit. In the top-ranked GO processes enriched in DM, the GO coverage was higher in DM than HC regardless of the GO process (Supplementary Fig. 6a). More than one-third (39.2%: 996 out of 2537) of the DM autoantibody targets were part of the 8 identified biological processes; 18.6% of them (186 out of 996) were also present in HC P20. DM processes involved structural elements including microtubule-based processes, actin-filament functions, cell junction organisation, extracellular structure organisation, cell morphogenesis in differentiation, and small GTPase mediated signal transduction (Fig. 5b). PTK2 and ROCK2 were shared between 7 of 8 functions, and, KIF14, NDEL1, SRGAP2, APP, PRKCZ and CLASP2 were shared between 6 of 8 functions (Fig. 5d). None of these proteins was observed in HC. The fact that the DM-specific and HC-specific proteins identified were robustly clustered in biological functions suggests non-random

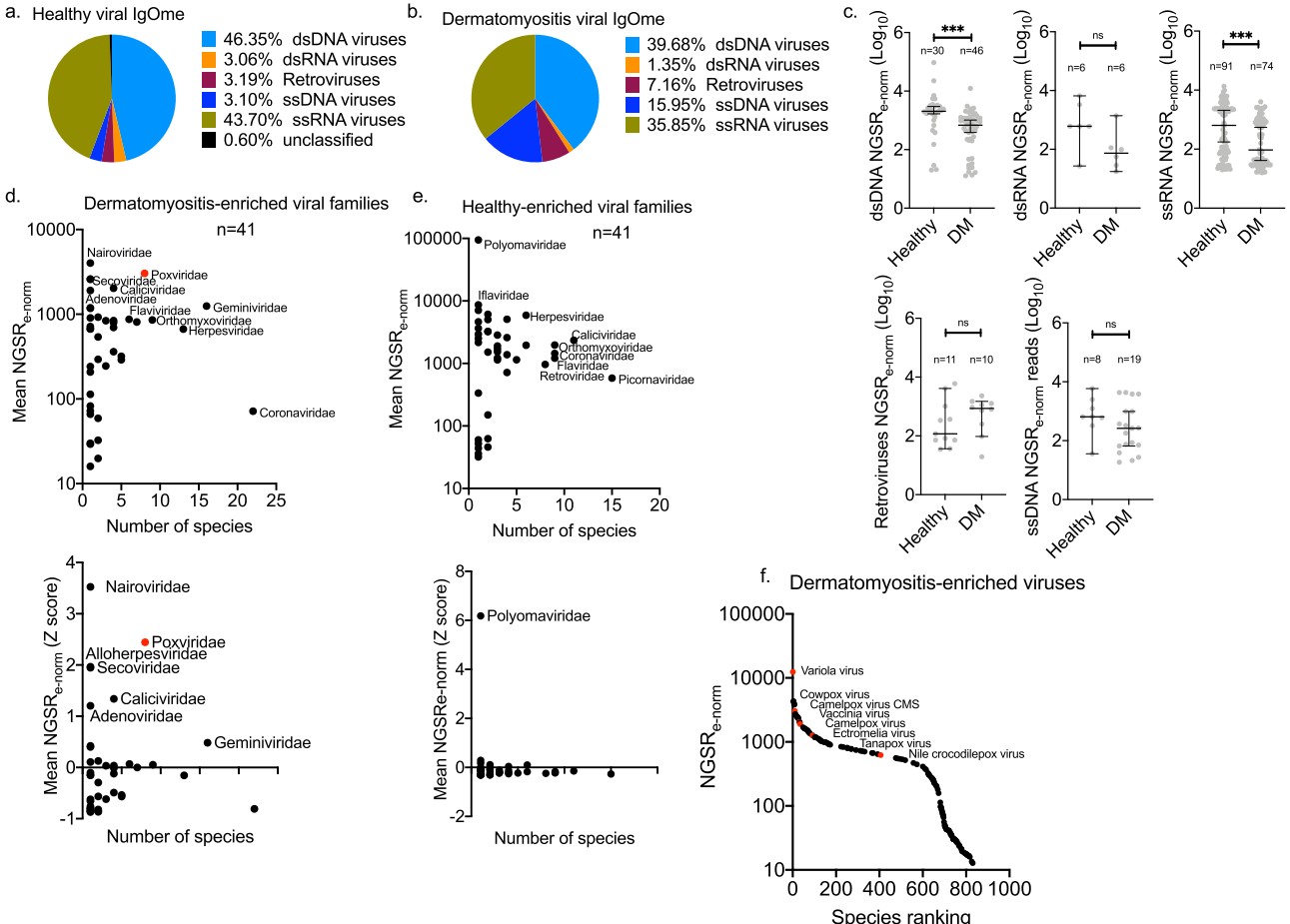

**Fig. 4 The dermatomyositis-associated viral IgOme.** Viral species were grouped based on genome type into six groups. The contribution of each viral group to the total NGSR$_{e-norm}$ output was evaluated in **a** healthy control (HC) P20 and **b** dermatomyositis (DM) P20. **c** Scatter plots of viral normalised abundance, mean NGSRe-norm, in DM and HC. Viruses are grouped based on their genome type. Scatter plots of viral family richness (number of species per family) and mean NGSRe-norm in: **d** DM, and **e** HC; in the upper panel NGSR$_{e-norm}$ values are depicted. In the lower panel, the NGSR$_{e-norm}$ values are Z transformed to identify viral targets with high NGSR$_{e-norm}$ (outliers). The Poxviridae viral family is depicted with red colour. **f** Scatter plot of species ranking based on NGSRe-norm in DM; species of the Poxviridae family are depicted with red colour. Differences were significant at the 0.05 level; * for $p < 0.05$, ** for $p < 0.01$, *** for $p < 0.001$. The median with 95% CI is annotated.

targeting of specific signalling pathways by autoantibodies. In DM, these processes share multiple autoantibody protein targets suggesting the presence of a DM-specific autoantibody-targeted proteome module (Supplementary Fig. 6b).

**Shared epitope sequences between *Variola* virus, Poxviridae, HIV and TRIM proteins**. We identified that antibodies against epitope sequences of the Poxviridae virus family were significantly enriched in DM, including *Variola* virus with the highest NGSR$_{e-norm}$ (Fig. 4f). Given that our DM patients were TRIM33 autoantibody-positive and due to the significant enrichment of TRIM proteins, IFNG and the IFN antiviral mechanism in our proteome data, we searched for potential links between *Variola*, other members of the Poxviridae family and TRIM proteins. Since molecular mimicry is a potential mechanism of autoantibody generation, we aligned the identified *Variola* virus and TRIM epitopes; the sequence annotation thresholds used in our bioinformatics pipeline guaranteed robust annotation of the epitope sequences. This would affect cross-kingdom epitope alignment since we maximised the phylogenetic distance between each microbial epitope and human proteins. To account for this, we started our epitope

sequence analysis with the widest possible diversity of identified *Variola* and TRIM epitopes, sacrificing specificity, i.e., epitopes with more than 80% match to *Variola* virus, and TRIM epitopes with an identity of more than 50%. Phylogenetic distances are shown in the circular cladogram in Fig. 6. We identified two different clades containing leaf nodes of TRIM and *Variola* epitopes of high similarity (branch lengths < 0.1). The first clade involved TRIM3 epitope "RIPDDVRRRPGC" and three additional epitopes "RI(Q)DDVRRRPGC", "RI(Q)DDV(H)RRPGC" and "RI(Q)DD(V)(S)RRPGC", which mapped to VARV GER58 hdlg 202 and VARV GUI69 005 202. The second clade contained TRIM3 epitope "SSHARYKSVRFS", and "SSHARYKSVRFS", "SSHARYKS(M)RFS", "SSHARYKSLRFS" "SSHARYKS(L)RF" and "SSHARYKSLRF(T)" of a *Variola* virus (unnamed protein product and viral DNA polymerase processivity factor). We reinstated the annotation specificity thresholds for TRIM and *Variola* epitopes to the default levels (strict, high specificity, maximisation of phylogenetic distances) (Supplementary Fig. 7a). Epitope sequence "RI(P)DDVRRRPGC" was retained and shared between TRIM3 (branch length: 0.0541) and VARV GER58 hdlg 202 (branch length: 0.0344). Epitope sequences "SSHARYKSLRFS" and "SSHARYKSLRF" were specific for *Variola* virus but no longer annotated as TRIM epitopes (Supplementary Figure 7a). We asked

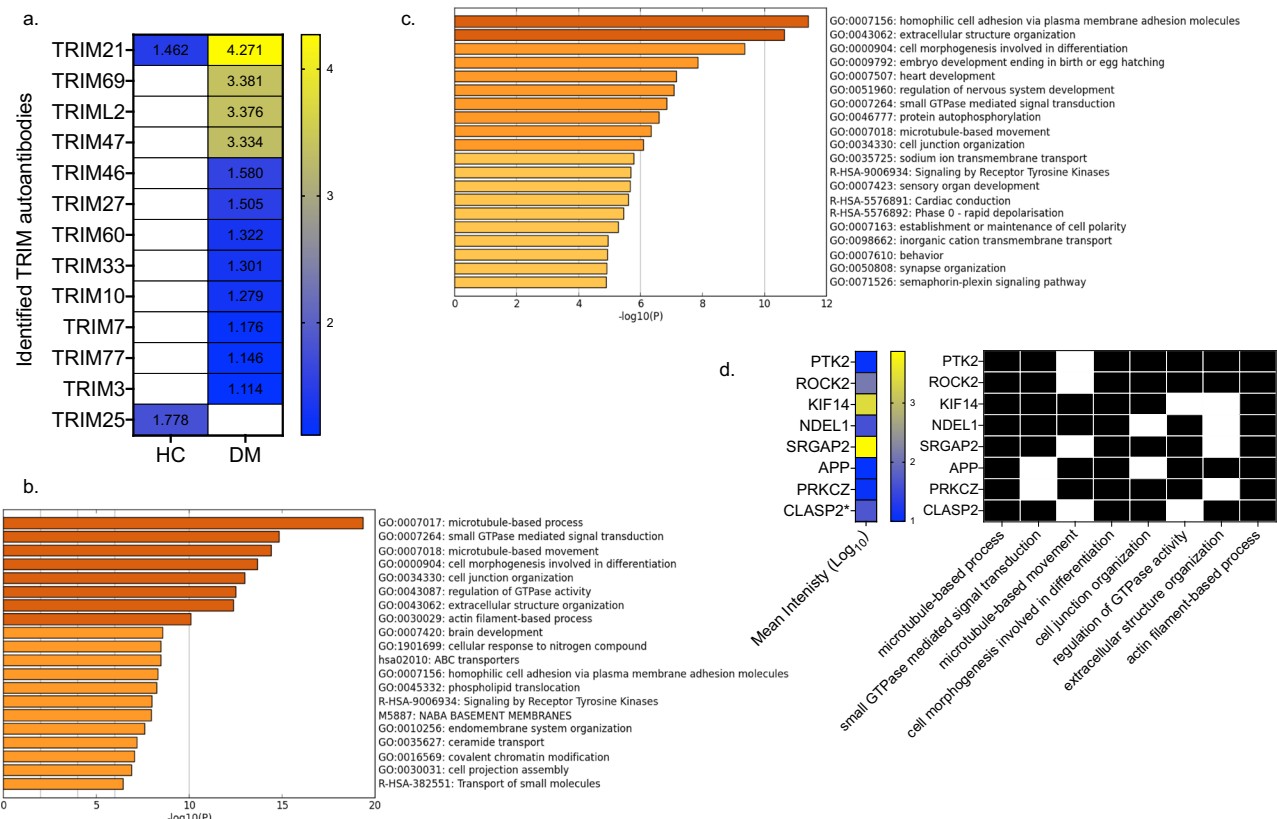

**Fig. 5 TRIM autoantibodies identified in Dermatomyositis. a** Heatmap of mean log₁₀ intensities of identified TRIM proteins in healthy control (HC) and dermatomyositis (DM). Gene ontology processes significantly enriched in **b** DM, and **c** HC. **d** Binary colour-coding of presence/absence of the autoantibody protein targets that were enriched in at least six out of eight DM-specific GO processes (presence: black, absence: white). The mean log₁₀ intensity of each protein is annotated in a colour coded heatmap (low: blue to high: yellow).

whether these epitope sequences are shared amongst different Poxviridae species that we identified in DM P20 (Fig. 4f). The three epitopes ("SSHARYKSLRFS", "SSHARYKSLRF" and "RIQDDVRRRPGC") were shared with high homology between *Variola, vaccinia, ectromelia, cowpox* and *camelpox* viruses (Supplementary Fig. 7b). We aligned the DM TRIM proteins that we identified to test the conservation level of the above epitopes in TRIMs (Fig. 7a). TRIM3 and TRIM33 shared high AA sequence similarity and seem to diverge from the rest of TRIMs across the region of interest (Fig. 7a).

Finally, we aligned all DM-enriched viral and TRIM epitopes (Supplementary Fig. 7c) and identified a third epitope "KHKGALGGGGNE" of TRIM47 which is shared by *Human immunodeficiency virus 1* (HIV-1) "KHKGALGGGG(N)E" and "KHKG(D)LGGGG(Y)E" shared by *Synechococcus phage syn9* "KHKG(D)LGGGG(Y)E" (Supplementary Figure 7d). The epitope was poorly conserved amongst the DM-specific TRIM proteins (Fig. 7b). Overall, we have identified two epitope sequences shared between *Variola* virus and TRIM3 and members of the Poxviridae family. A third epitope shared between TRIM47, *HIV-1* and *Syn9* was also observed. These findings suggest that molecular mimicry events, at least amongst these specific viral and TRIM epitopes, is a potential mechanism of pathogenesis in DM.

## Discussion

This is the first study to investigate the accumulated microbial and autoantigen antibody repertoire in adult–onset DM patients using an un-targeted high-throughput approach to identify

immunogenic epitopes. In DM, antibodies were characterised by a higher number of epitopes per microbial species and against a wider microbial repertoire. The effect of sample size in our measurements suggests increased microbial exposure and inter-personal variability in DM compared to the HC group, leading to the expansion of the DM-specific antibody microbial signature identified upon screening additional samples. Although our observations provide a static snapshot of the IgOme, the accumulation of antibodies takes place over an individual's lifetime. Thus, our data suggest that DM as a clinical entity is characterised by diverse microbial exposure.

Antibodies recognising viruses were over-represented amongst the microbial species, with the highest abundance in DM and HC. The competitive design of the epitope enrichment process suggests that viral exposure has a role in DM, additional potential differentiating cellular microbes will be further investigated in the future. Enriched antibodies targeting virus families in DM included Coronaviridae (primarily *SARS* coronaviruses), Ortho-myxoviridae (primarily *Influenza A* viruses), Herpesviridae (including A and B groups), Geminiviridae plant viruses probably due to exposure via the gastrointestinal tract, and the Poxviridae Smallpox (*Variola*) and *vaccinia* viruses. Epitopes against Pox-viridae species had the highest abundance. The smallpox virus *Variola* was eradicated by the introduction of the *vaccinia virus* (VACV)-based vaccine[14]. The last worldwide vaccination pro-gramme was in 1967[14], and smallpox immunity persists for many years[15], therefore given the age of our donors it is likely that the DM associated antibodies to Poxviridae were raised through the vaccination process. This is supported by the observation of *VACV* antibodies in the healthy sample without

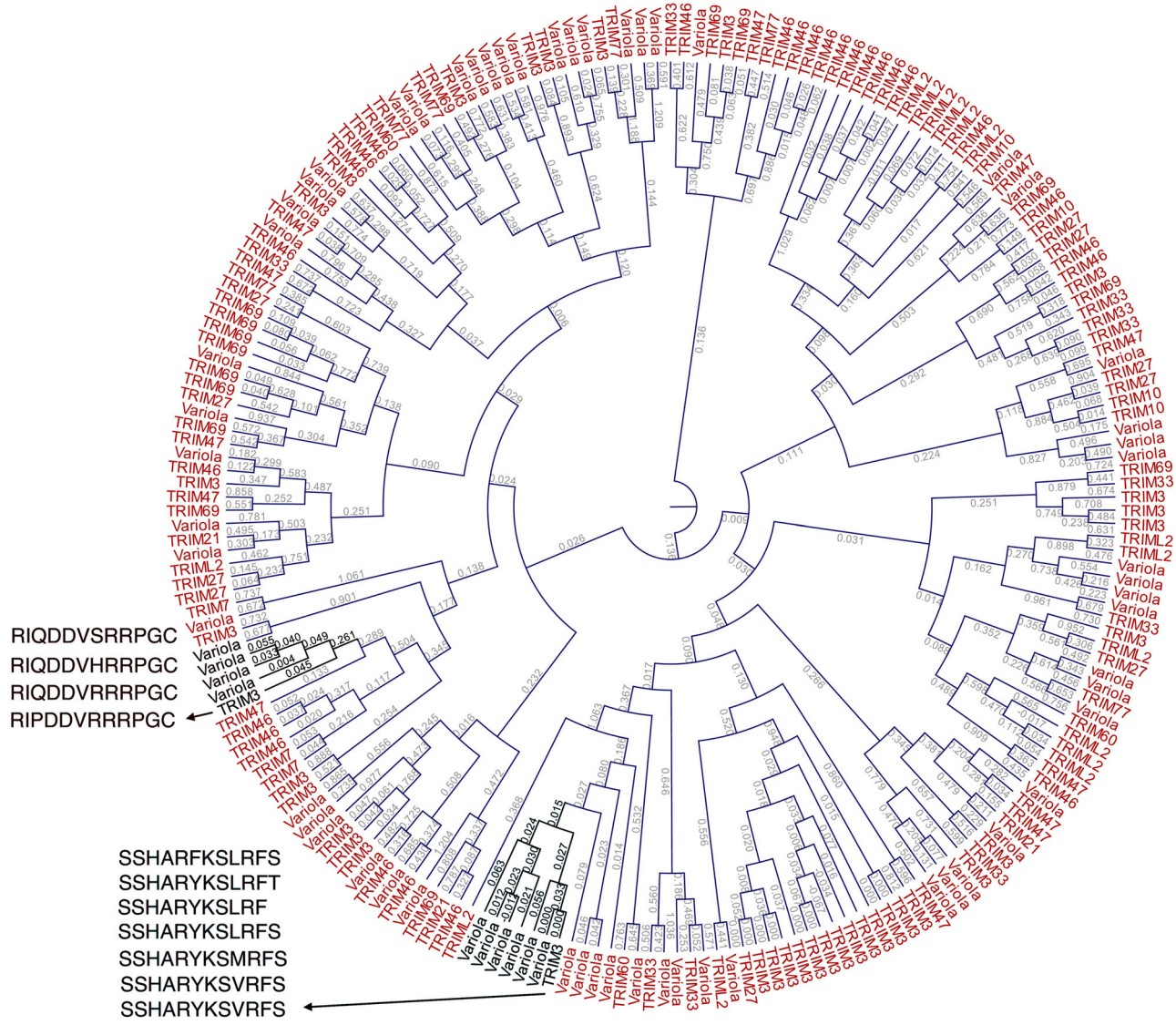

**Fig. 6 Circular cladogram of phylogenetic distances between *Variola* virus and TRIM epitopes identified in dermatomyositis.** The DM-specific epitope sequences identified in TRIM3 and *Variola* virus had the highest similarity and are annotated next to the cladogram clades. Branch lengths represent the phylogenetic distance (Kimura protein distance).

being enriched. Notably, antibodies against *vaccinia/Variola* virus are often observed in non-immunised infants of vaccinated mothers suggesting a continuum of *VACV* protection even in the post-eradication era[16,17]. Antibodies against many other viral species capable of directly infecting muscle tissue were enriched in DM, including *human immunodeficiency virus 1 (HIV), human papillomavirus virus (HPV), hepatitis C and B (HBV), enterovirus A71* and foot-and-mouth disease virus, and human Adenovirus and Rotaviruses which have been identified in sporadic cases of infectious myositis[18–20]. DM plasma also was enriched in antibodies against viruses reported to affect the musculature indirectly via immune mechanisms, such as Influenza, Enteroviruses, HIV, SARS-Coronavirus, Herpes viruses and *Parvovirus B19*, and which have been implicated in the pathogenesis of polymyositis and DM[21,22]. Case studies report the development of DM after vaccination against some of these viruses, notably *HBV, HPV,* influenza and *VACV*[23]. Overall, the DM-specific antibody signature is directed towards viruses that can directly infect muscle tissue and/or indirectly sabotage immune homoeostasis. The respiratory and gastrointestinal systems as the primary physiological targets of these viruses agree with the high frequency of this type of infection in patients with juvenile- and adult-onset DM[24,25]. Moreover, the age-range of juvenile and adult-onset DM coincides with the epidemiological peaks of respiratory infections during life[26].

In DM, we identified autoantibodies against >2500 human proteins; about 14% of the current human proteome (UniProtKB/ Swiss-Prot database, last modified 29 July 2019). This was double the proportion compared to the HC-enriched proteome, with only 13% of autoantibodies found in both DM and HC, suggesting accumulation of autoantibodies in DM against a wider group of proteins. Three times the number of autoantibody targeted proteins were regulated by type I and II interferons in DM than in HC. This underscores the relevance of interferons in DM[27,28] and indicates the potential of interferon blockade as a therapy. Identification with high confidence of specific biological processes enriched in both DM and HC suggests an organisational structure within the targets of these accumulated autoantibodies. This is consistent with an infection (exposure)-driven pathological autoantibody-producing mechanism which

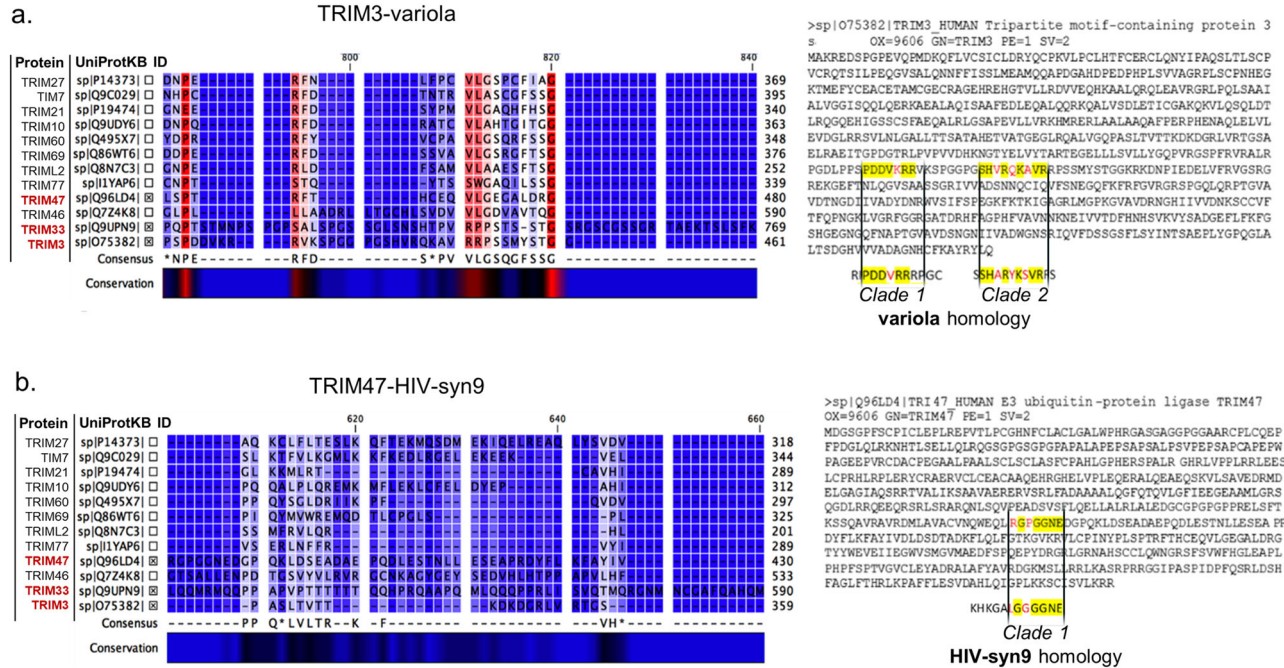

**Fig. 7 Partial amino acid alignments of the DM-associated TRIM proteins that share amino acid similarity with viral species.** Alignment of all identified TRIM proteins in DM (Fig. 5a); TRIM3 is located on the bottom of the alignments plots. Above TRIM3 (sp|O75382) are TRIM33 (sp|Q9UPN9) and TRIM47 (sp|Q96LD4); cross-marked. Conservation level is presented as a colour gradient (high; red, low; blue). **a** Partial alignment of the region where the TRIM3-*Variola* high similarity epitopes were identified. The TRIM3 protein sequence is presented next to the alignment plot. Yellow: the two TRIM3 epitopes with high amino acid (AA) similarity to *Variola* and poxviruses (Fig. 6). Black font: identical AAs between TRIM3 and *Variola* virus. Red font; different AAs between TRIM3 and *Variola* virus. Both epitopes are located only 6 AAs apart (relative to TRIM3 protein sequence). TRIM3 presents high similarity with TRIM33 in respect to the specific AA epitopes. **b** Partial alignment of the region where the TRIM47-*HIV-syn9* high similarity epitope was identified. The TRIM47 protein sequence is presented next to the alignment plot. Yellow: the TRIM47 epitopes with high AA similarity to HIV and *syn9 phage* (Supplementary Fig. 7d). Black font: identical AAs amongst TRIM47 and the two viruses. Red font; different AAs amongst TRIM47 and the two viruses. The epitope is poorly conserved amongst the DM-specific TRIM proteins.

progressively expands its repertoire (antigens) under the chronic burden of accumulated stimuli[29–32]. Biological processes enriched in DM are orientated around the cytoskeletal organisation, microtubule movement, actin filaments and cell junctions. Eight identified protein targets participated in most of these processes indicating a key role in the regulation of multiple signalling pathways. Enrichment of antibodies against proteins regulating focal cell-cell adhesion in DM is interesting, since disruption of these structures in epithelial cells facilitates the spread of viruses, by allowing their release from the basal to the apical surface or the external environment[33,34]. Some of the DM associated viruses described here manipulate junctional proteins namely Adenoviridae, Flaviridae, Herpesviridae, Retroviridae, Paramyxoviridae and Picornaviridae which provides a functional link between DM and increased viral exposure[35–37].

We identified autoantibodies against TRIM33 in DM plasma, confirming the anti-TIF1 positive profile of our patients. Furthermore, we identified autoantibodies against 11 other TRIM proteins in DM, of which only TRIM21 was identified in HC but with much lower abundance. TRIM21 (Ro52) is a known autoantigen in IIM and related autoimmune disorders and acts as the highest-affinity Fc receptor in humans. TRIM21 binds cytosolic antibodies bound to non-enveloped viral pathogens, to trigger antibody-dependent intracellular neutralisation and proteasomal degradation of the immune complex[38,39]. The autoantibodies we identified against TRIM proteins, other than TRIM33 and TRIM21, have not been observed previously in IIM. TRIM69 inhibits vesicular stomatitis virus transcription and mediates poly-ubiquitination and degradation of dengue virus non-

structural protein 3[40,41]. TRIMs also regulate antiviral pathways indirectly by mediating innate immunity, influencing the transcription of TI-IFNs, interferon-stimulated genes and pro-inflammatory cytokines[6]. The TRIM protein family expanded very rapidly in evolution, coinciding with the development of the adaptive immune system, suggesting that TRIMs evolved to fine-tune interactions between the innate and adaptive immune systems[5,39]. The deregulated activity of one-third of TRIM proteins has been associated with the development of human cancer[42]. TRIM47 and TRIM27 have an oncogenic role in colorectal, prostate, oesophageal, ovarian and non-small cell lung cancer, promoting proliferation and metastasis[43–45]. Conversely, TRIM3 inhibits the growth of liver, colorectal and gastric cancer[46]. Identification of autoantibodies against cancer-associated TRIMs accords with the strong temporal association between myositis and development of malignancies in adult-onset anti-TIF1γ-positive DM[4], consistent with the high proportion of cancer-associated myositis cases in the current study (45%), including ovarian, lung and hepatic cancer. Overall, the expanded targeting of TRIM proteins observed in DM plasma supports the role of the TRIM RING-type E3 ubiquitin ligase subfamily as powerful regulators of the immune system through post-translational modification[6,47].

Since Poxviridae epitopes were enriched in DM, we investigated a potential link between poxviruses and TRIM proteins. We identified epitope sequences with the high similarity between TRIM3 and *Variola* and other members of the Poxviridae family. TRIM47 had epitopes in common with *HIV1* and *Synechococcus phage*. These high similarity epitopes are in the C-terminus of the

TRIM proteins; this region is under evolutionary positive selection and determines ligand binding specificity, function and subcellular localisation, and may be associated with the ability to restrict viral infection, particularly retroviruses[6]. This finding of high similarity epitope sequences between specific viral and TRIM epitopes suggests that molecular mimicry is a potential mechanism of pathogenesis in DM.

Finding antibodies against a human protein does not mean that this was the antigen against which the antibody was raised, as pathogen-derived proteins can mimic endogenous epitopes[31,48]. This might explain inconsistencies between the universal cellular distribution of specific antigens and the primary skeletal muscle target of the disease[48]. We observed that DM-specific TRIM epitopes did not form a separate cluster but were dispersed throughout the viral epitope phylogenetic tree, suggesting that accumulated antibodies against TRIM proteins might be the product of a primal molecular mimicry event and subsequent epitope spreading rather than increased TRIM protein production. This is supported by the observation that gene expression levels of autoantigens in myositis muscle biopsies do not correlate with the levels of circulating autoantibodies recognising each cognate endogenous autoantigen; instead, they are associated with expression of muscle regeneration markers[49].

We propose a model whereby DM pathogenesis depends on viral exposure patterns, which start early in life. The physiological transmission of viruses to humans is mainly via the respiratory and gastrointestinal systems, with non-physiological presentation through immunisation. The link between DM and viruses is complex but striking in our data since there is systematic enrichment for human proteome modules that are directly exploited by, or respond to, most of the viruses we identified. We propose that an initial viral infection or virus delivery system presents the main antigen that induces antigen-specific antibody production in a predisposed immune environment. Since TRIM proteins share epitope homology with multiple viral species, these antibodies can also bind to TRIMs and homologous proteins. This process is influenced by the innate immune system status, antibody populations at the site of infection (local antibodies), the virus-specific serum antibody concentration (which usually peaks several weeks post-infection), and history of exposure to the same or related viral strains (and epitopes)[50]. Since HLA Class II molecules are regulators of the adaptive immune response to antigenic challenge through T cell repertoire selection, antibody production to viral antigens also may be mediated by HLA type. We previously identified that adult-onset anti-TIF1-positive DM is associated with HLA-DQB1*02:02[51]; in this study 6/15 (40%) DM and 3/17 (18%) HC samples with available data are heterozygous for HLA-DQB1*02:02. In myositis, correlation of autoantigen and muscle regeneration marker expression[49] suggests that after the initial viral presentation, increased concentrations of muscle proteins may be sufficient even in the absence of viral proteins to invoke periodic rises of autoantibodies. The continuous activation of innate immunity in DM[28,49,52,53] can also potentiate autoimmunity through chronic immune-mediated tissue damage resulting in autoantigen release without the need for specific activation of auto-reactive T cells by a microbial mimic[31]. Following initial viral exposure, there is a high chance that this effect will spread amongst related proteins within specific signalling pathways of the initial hit (since protein homology is related to function). In this model, accumulated autoantibodies against proteins that participate in specific biological processes would be observed in autoantibody-positive myositis subgroups. In HC the protein coverage of the GO processes significantly decreases suggesting more random targeting of human proteins by autoantibodies, opposite to that observed in DM, in support of our hypothesis.

The use of pooled plasma is a limitation in our study since we cannot deconvolute the data into individual patients. Using equal amounts of isolated immunoglobulins from all samples and subsequent normalisation ensures that our results are not heavily skewed towards the profile of a single sample. Moreover, the enrichment process against healthy adults allows investigation of DM as a disease system capturing the breadth of microbial antibody and autoantibody accumulation against a large number of linear epitopes. Apart from antibodies against TIF1, a shared feature of patients in this study, we do not anticipate that our findings will be equally distributed among patients. If the shared feature in DM is a molecular mimicry event combined with epitope spreading, then the stochastic and dynamic nature of microbial exposure, genetic predisposition and immunity would not support a single pathotype. Thus, our study provides a DM-associated "map" of potential routes to disease. For the future, we recommend: clinical records are kept of microbial exposure, infections history including pathogen, severity scores, number of hospitalisations and vaccination history; information not available in this study. In juvenile DM, records could include maternal infection and vaccination history to account for acquired transplacental immunity.

Lifelong exposure to viruses contributes not only to the accumulation of virus-specific antibodies and protection but the generation of autoantibodies, which may reach a critical mass and induce disease as a result of mild or non-symptomatic infection. Molecular mimicry and epitope spreading may play a role, raising questions about the extent and sequence of events. This suggests that autoantibodies identified in DM to date might only be the tip of the iceberg.

## Methods

**Study cohort.** Plasma samples were collected from anti-TIF1 positive adult-onset dermatomyositis (DM) patients through the UK Myositis Network, as described previously[51] (Supplementary Table 1). All individuals fulfilled definite or probable Bohan and Peter classification criteria for dermatomyositis. Anti-TIF1 autoantibody positivity was identified by immunoprecipitation, and confirmed by ELISA, as described previously[51]. Samples were selected at random and based on plasma availability. Gender and age-matched (at time of sample collection) healthy controls (HC) were randomly selected from the University of Manchester Longitudinal Study of Cognition in Normal Healthy Old Age cohort[54]. All samples were collected with relevant research ethics committee approval (MREC 98//8/86 North West Haydock Research Ethics Committee for UKMyoNet and UREC 02225 and UREC4 2017-1256-2489 for healthy control cohort). Study participants provided written informed consent. Clinical, demographic and experimental information is presented in Supplementary Table 1.

**Experimental protocol.** We implemented the "SARA" pipeline (manuscript in preparation). SARA comprises a comprehensive workflow that integrates molecular biology peptide display and antibody epitope signature enrichment through competitive bio-panning and high-throughput DNA NGS, alongside in-house computational scripts to reverse engineer high-resolution antibody epitope signatures that reflect original in vivo antibodies against epitopes present in patient sera. SARA was applied to predict the identity and abundance of antibody epitope repertoires enriched in plasma from anti-TIF1 autoantibody-positive DM patients versus plasma from matched HC (see Supplementary Methods). This pipeline provides a digital triage of infectious organism epitopes and autoantibodies predicted to be uniquely present or highly enriched in each plasma pool. All informatics analysis was carried out using R[55] 3.2–3.6 and Python[56] 3.5. We briefly summarise the seven SARA pipeline modules (M1–M7) below in turn.

### SARA pipeline

*Plasma total immunoglobulin purification: M1.* Total immunoglobulins IgA, IgG and IgM were purified from twenty anti-TIF1 positive adult-onset DM and twenty HC plasma by AdamTech Total Ig Extraction kits (Pesac, FR). For the sample-pools of 10 (P10), 12 μg of isolated Ig per donor were used for a total of 120 μg. For the sample-pools of 20 (P20), 6 μg of isolated Ig per donor were used for a total of 120 μg. Ig yields and purity were assessed by NanoDrop spectrophotometry and polyacrylamide gel electrophoresis.

*Competitive biopanning: M2.* Separate purified Ig pools from DM and HC were used for competitive bio-panning with the FliTrx™ random 12 AA peptide surface

display system[57]. Briefly, we first conducted a pre-panning stage to incubate separate DM and HC immobilised Ig pools with induced FliTrx[TM] E. coli cells[57] in order to sequester expressed epitopes relevant to the specific plasma cohort. Unbound bacteria per cohort were further incubated with the alternative immobilised Ig pools for the main panning stages (cross-panning). Tethered bacteria were eluted and expanded to repeat the biopanning process five times. OD600 was measured after each round of biopanning to ensure comparable efficiency between DM and HC (Supplementary Figure 3e). After competitive biopanning immobilised bacteria were expanded and polyvalent plasmid cohorts were purified (Maxiprep, Qiagen, UK).

Epitope NGS: M3, NGS data processing: M4. Polyvalent plasmid variance regions were PCR amplified and gel-purified. PCR products were validated by Sanger sequencing (Supplementary Figure 3f), and NanoDrop. The DNA primer sequences are FliTrx[TM] Forward: 5′-ATTCACCTGACTGACGAC-3′, FliTrx[TM] Reverse: 5′-CCCTGATATTCGTCAGCG-3′. Multiplexed DNA fragments were sequenced on the NextSeq 500 platform (Illumina, UK). We retrieved ≈24 million and ≈36 million paired-end FASTQ reads for HC and DM, respectively (Fig. 1). The 36-bp variance region sequences were translated with respect to the reading frame. <14% of DM and <13% of HC comprised premature stop codons, indicative of immunologically relevant bio-panning enrichment (Supplementary Fig. 1a–d). Non-specific HTS sequence noise from residual bio-panning solution was controlled for by our 10-σ-99 noise floor. This retains all expressed epitopes within each respective HC or DM pool that surpass a threshold of 10 standard deviations above the lowest 99% distinct peptide sequence enrichment when ranked by reading count (further described in Supplementary Fig. 1a–d).

Epitope signature set analysis: M5. Distinct epitope signature set analysis presented unique and common AA epitope sequence sets with associated NGS enrichment scores. Minimum thresholds of 10AA sequence length and the 10-σ-99 NGS noise filter provided sensitive annotation power while minimising annotation noise (Supplementary Fig. 1d–f). Cleaned distinct epitope sequences were collated. Unique epitope pools were determined by the symmetric difference between the HC and DM pools (i.e., specific epitope sequences uniquely found in HC or in DM patients). The intersect between HC and DM pools was probed and a minimum fivefold changes threshold applied to partition relevant epitope collections to supplement the HC or DM pools. These epitope collections were utilised in the downstream microbiome and autoantibody annotations. >8.6 × 10^6, and >4.7 × 10^6 distinct epitopes with a minimum length of 6AA length were retrieved for DM and HC pools, respectively. The highest NGS read counts achieved were ≈111,000 (DM) and ≈88,000 (HC) (Fig. 1).

Epitope annotation and rank scoring: M6. Totally, 15,522 DM-associated and 4817 HC-associated unique epitopes were retrieved for DM patients. Only 0.65% of the DM pool comprised peptide sequences common to HC. Of these epitopes common by sequence, the DM enrichment score ranged between 5-fold and 546-fold the healthy control score (Fig. 1) (Supplementary Fig. 1a–f). Epitopes were annotated by our modified BLASTp[58] approach at 100% minimum identity (Supplementary Fig. 1g–i). This facilitates short AA-sequence inputs and resolves protein annotation confidence scores per epitope with statistical control of epitope-protein annotation confidence, intra-protein dis-contiguous sequence matches, redundancy of protein isoforms and common sequence matches, single-epitope multiprotein parsing, and set analysis of resultant patient cohort organisms lists. These were tuned per each epitope's NGS read count. 6.745 × 10^6 (DM patients) and 2.250 × 10^6 (HC) total protein annotations were retrieved (Fig. 1).

Phylogenetic and taxonomic analysis: M7. Our scoring system generates aggregate NGS read count enrichment and annotation stringency scores. We retrieved every known upstream taxonomic ranking level for all microbe IDs and anchored these via NCBI REST and the Taxonomy repository[59–61]. Phylogenetic trees were generated by PhyloT biobtye[62] and microbial annotations were assessed by NGS enrichment, epitope mapping confidence, annotation scores, taxonomic ranks and phylogenetic clustering (Supplementary Fig. 1g–i). We utilised the interactive Tree Of Life iTOL[63,64] to taxonomically cluster microbial agents for the DM or HC pools with incorporated annotation layers representing the NGS enrichment and annotation scores (Supplementary Fig. 1i). Totally, 9111 distinct and 6202 unique species were annotated for DM patients (4994 distinct and 2085 species were annotated for HC) (Fig. 1).

**Human protein autoantibody digital triage.** We identified any autoantibodies against human proteins that may be present. Annotation scores occupy unit interval space [0,1] as surrogate significance values. Protein cohorts were inspected by Metscape[65], KEGG[66], UniProt[67] and the Human Protein Atlas[68], to explore functional predictions based on aberrant autoimmune targeting. Totally, 435 (DM) and 407 (HC) enriched human biomarkers from autoantibodies were retrieved (Fig. 1).

**Phylogenetic analysis of epitope sequences.** Alignments of human and viral epitope sequences were performed with a gap open cost of 10.0 and a gap extension

cost of 1.0. Tree (circular and radial) construction was performed using Neighbour-joining and Kimura protein distance measure with bootstrap analysis (100 replicates). The analysis was performed in CLC Genomics workbench 12.

**Statistics and reproducibility.** Continuous variables were $\log_{10}$ transformed and tested for normal distribution using the Shapiro–Wilk test. Pairwise quantitative differences between two groups were tested using two-tailed Mann Whitney (Wilcoxon rank-sum test) for variables not following a normal distribution and unpaired $t$ test for variables with a normal distribution. $F$ test was used to test for equal variance. The Kruskal–Wallis nonparametric test was used to compare three or more groups; Dunn's multiple comparisons test was used to compare pairwise differences in the sum of ranks with the expected average difference (based on the number of groups and sample size per group). $Z$ score transformation was used to specify the location of each observation within a distribution. Statistical analyses were performed with GraphPad Prism 8.0 (GraphPad Software, Inc., San Diego, CA)[69].

**Reporting summary.** Further information on research design is available in the Nature Research Reporting Summary linked to this article.

## Data availability
Raw sequencing data have been deposited in DRYAD (https://doi.org/10.5061/dryad.gmsbcc2mb). Human proteins and microbial annotations can be found in DRYAD (https://doi.org/10.5061/dryad.gmsbcc2mb). Source data can be found in Supplementary Data 1. All other data are available from the corresponding author on reasonable request.

## Code availability
The code supporting the current study have not been deposited in a public repository but will be deposited following the submission of our methods manuscript (in preparation). In the intervening period, these will be available from the corresponding author on request.

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

## Acknowledgements

This study was supported by a research grant from The Myositis Association. T.D.J.W. was supported by a research grant from Children with Cancer and The Caring Cancer Trust. X.H. was supported by grants from The Humane Research Trust and work in the Viral Oncology Labs was supported by grants from the Cancer Prevention Research Trust. H.C. and J.L. were supported by the Medical Research Council (MR/N003322/1). H.C. was supported by the NIHR Manchester Biomedical Research Centre Funding Scheme. The views expressed in this publication are those of the authors and not necessarily those of the NHS, the National Institute for Health Research or the Department of Health.

## Author contributions

Conceptualisation: I.H., L.H., T.D.J.W., X.H., S.M., W.E.R.O. and J.A.L.; Methodology: I.H., L.H., T.D.J.W., X.H. and S.M.; Software: T.D.J.W. and X.H.; Investigation: S.M., T.D.J.W., X.H. and J.O.'S.; Resources: I.H., L.H., N.P. and A.P.; Writing—Original Draft: S.M., T.D.J.W. and J.A.L.; Writing—Review & Editing: S.M., T.D.J.W., J.A.L., I.H., L.H., X.H., W.E.R.O., H.C., J.O.'S., N.P. and A.P.; Visualisation: S.M. and T.D.J.W.; Supervision: I.H., L.H. and J.A.L.; Funding Acquisition: J.A.L., I.H., W.E.R.O. and H.C.

## Competing interests

The authors declare no competing interests.
