## [Peer Review File · Communications Biology]

Reviewers' comments:

Reviewer #1 (Remarks to the Author):

Early onset dermatomyositis (DM) is an autoimmune disease that can be associated with auto-antibodies against TRIM33. In the current manuscript, Megremis et al utilize "Sara", a platform for identifying epitopes that are bound by the individual serum antibodies, in healthy controls and dermatomyositis patients, in order to identify epitopes (belonging either to microbes or self-proteins) that are specific or enriched in patients vs controls. The authors identify a large set of epitopes, many of which are enriched in one of the two groups. Particularly viral epitopes, TRIM proteins and type I interferon-induced proteins are found more often or uniquely in DM patients, and several epitopes share homology between TRIM proteins and poxviruses. The authors conclude that viral infections might be at the heart of the autoimmune disease

The role of microbial infection in the aetiology of autoimmune disorders is intriguing and very much worth studying. The current manuscript provides a large number of possible leads that might help to unravel the cause and/or pathogenesis of DM, which is of great clinical importance and can be extrapolated to multiple other autoimmune diseases. I believe the used method is state-of-the-art and has the potential to identify, in an unbiased and high-throughput manner, relevant epitopes in various autoimmune diseases. However, I am somewhat worried about the conclusions that can be drawn from the data as presented by the authors, particularly whether the identification of an antigen that putatively binds a serum antibody actually means that the person actually was infected/exposed to the organism expressing the antigen, or that there might be a certain level of "noise". I have a number of questions/comments that may improve the current manuscript:

1. The viral epitopes found suggest that the patients have been exposed to Variola virus (smallpox), which is eradicated but used in vaccines until 1967. Are there no patients or controls born after 1967?
2. In addition to Variola, the DM patients are enriched in antibodies binding epitopes from a pretty aggressive set of viruses, including SARS, HIV, foot and mouth disease virus and HBV/HCV. Is this realistic? There must be a rich clinical history present when patients have such diseases. Do rare and highly pathogenic viruses more often infect future DM patients than healthy controls? Or is it possible that a number of the identified epitopes share accidental homology to these viruses?
3. The authors explain the presence of antibodies against Geminiviridae plant viruses through exposure to them via food. Why are they then enriched in the DM group? How is the presence of antigens from bacteriophages explained?
4. As the authors also state, the use of pooled plasma is a limitation of this study design. When pooling 10 or 20 sera from DM or HC, how do the authors exclude for the possibility that one patient screws the entire pool, creating the artificial appearance of 'group-specific' epitopes?
5. When looking for shared epitopes between viruses specific for DM and TRIM proteins, it might be informative to compare that to similar analyses done on the HC pool, or include other negative controls; this would eliminate the feeling of cherry picking.
6. Figure 1 and 2 are technical and provide a good overview of the results for the analyses. However, they could be combined or partially added to a supplemental figure.
7. Figures S2A and B look impressive but are quite useless at this resolution.

8. Line 170: "The over-representation of viral compared to cellular microbial species in the top-ranked microbes enriched in DM and HC suggests an important role of virus exposure in the environment-host immune cross-talk in both healthy and disease states, and implies that viruses play an important role in the pathology of DM." Why do the authors draw the last conclusion?

Reviewer #2 (Remarks to the Author):

In the manuscript "Microbial and autoantibody immunogenic repertoires in TIF1 γ autoantibody positive dermatomyositis" Spyridon Megremis, et al, report the anti-microbial and anti-autoantigen antibody repertoire in patients with adult-onset dermatomyositis that are sero-positive for TIF1 γ (TRIM33) autoantibodies. The authors use a very interesting high throughput approach to identify the IgOme against microbial and human epitopes, with an increased diversity of antibodies directed to microbial epitopes, particularly viruses of the Poxviridae family. Further, the authors indicate that the autoantibodies identified are directed against a large portion of the human proteome, and emphasize on TRIM proteins which have homology with viruses, including poxviruses. Thus, they claim molecular mimicry and epitope spreading as having a potential significant role in the pathogenesis of dermatomyositis. However, I have a few concerns that need to be addressed to strengthen this report.

Title.

The manuscript reports the repertoire of epitopes recognized for antibodies in pooled plasma of dermatomyositis patients as compared with pooled plasma from healthy controls. With this in mind, to title the manuscript as "the microbial and autoantibody immunogenic repertoire", seems somehow confusing. It needs to be clear that what is reported is the repertoire of antibodies recognizing these antigens.

This holds true for the entire manuscript in which most of the time the text reads as if microbiome techniques were used.

Results.

1. Subheading: Higher microbial diversity in dermatomyositis. This report does not report on microbial diversity, but on the antibody repertoire. Thus the data does not demonstrate the presence of a stably-enriched microbial component in DM, especially when one of the main claims is the potential molecular mimicry and epitope spreading.

2. It is important to state the rationale for the use of two different pools in each group.

3. Poxviruses are tightly linked with dermatomyositis. Antibodies directed against dsDNA viruses, group in which poxviruses are included, are contained in a higher proportion in the HC pools. However, when each pool was individually evaluated for antibodies against each family, Poxviridae was the only family that ranked highly in the DM pool. Amongst these the highest was variola virus. As mentioned later in the manuscript, the likelihood of these patients having been vaccinated is high, however, it is important to include the actual vaccination status of these patients and controls (as best possible) in the corresponding table.

4. While I assume the data will be made publicly available, it is important to include the unsupervised identification of human protein targets for reference (maybe as a supplementary figure). Where those TRIM and IFN-regulated proteins rank? Are they in among the most significant hits? Is there a differential signature between the two pools?

5. The authors identify a series of epitopes that are shared between poxviruses and TRIM3 in DM. I think that demonstration of the capability of these plasma samples to recognize TRIM3 (IP, WB, ELISA?) would shed insight on the relevance of these antibodies in the etipopathogenesis. Also, as indicated in the discussion (line 382), TRIM3 is considered as an inhibitor of liver, colorectal and gastric cancer; however, only one of the patients included in the study have DM-associated liver

- cancer associated. What are the potential implications of this seemingly discrepancy? Please discuss.
6. Pooling plasma from patients with DM-associated cancer and those cancer-free, may be skewing the results toward a different autoantibodies repertoire, thus I believe comparison of these two groups is crucial.
 7. The authors suggest the accumulation of antibodies against proteins that strongly contribute to IFN gamma signaling and the broader antiviral mechanism by IFN-regulated proteins in DM patients (lines 243-245). Do these patients have more frequent viral infections? How can this be aligned with what seems to be a potential ability to mount better antiviral responses?

Discussion

1. As mentioned before, it is important to make it clear that the data shown is on the antibodies against microbial epitopes and human proteins; no microbiome studies were performed. For instance, Viruses were not over-represented (Line 321), but antibodies recognizing viruses were over-represented; Enriched families in DM included... (line 323); DM plasma also was enriched in viruses reported to affect... (line 335); etc. Please be sure that this is clarified in the entire manuscript.
2. The authors acknowledge the limitation of using pooled plasma. In this regard, it is very important to differentiate between patients with DM-associated cancer, which they would not be able to accomplish with the strategy undertaken.

Methods

1. There are no epitopes in pooled sera, but antibodies against those epitopes (Line 479)
2. There were no pooled sera but plasma (line 483).

Figures

1. Figure 1. This figure indicates work with pooled sera, but all the text indicates pooled plasma; which was used?
2. Figure 3 (legend). Ranking of identified microbial species... This manuscript does not identify microbial species, but the epitopes to which plasma antibodies are directed against.

Supplementary table 1

1. The column for ethnicity contains mixed information, i.e. ethnicity (White British), and race (Caucasian). This needs to be corrected, or eliminated the authors need to consider that 'oriental' is neither a race nor an ethnicity, thus this term needs to be replaced for the correct one. However, the authors need to explain what the importance of this information is, since all the patients and controls, but one of the patients seem to be of the same race? In fact, one wonders whether the inclusion of only one patient from a different racial group is adequate, or represents a further limitation of the study in terms of both lack of diversity or the potential of skewing the IgOme in a way that is not representative of the racial majority included; please discuss.
2. Please clarify the cancer associated myositis definition in the table legend. Is this defined as cancer that was diagnosed within three years after the onset of myositis?
3. Considering the importance of TRIM proteins as oncogenic proteins (TRIM27, TRIM47) or tumor suppressors (TRIM3), and the fact that the healthy control pools did not have antibodies against epitopes in these proteins, it would be very informative to establish the malignancy status in the set of healthy subjects.

Supplementary tables 2 and 3

1. Please clarify that is not richness of viral families but the antibodies against them.

Reviewer #3 (Remarks to the Author):

I have read with interest the manuscript titled, "Microbial and autoantibody immunogenic repertoires in TIF1 γ autoantibody positive dermatomyositis". The paper focuses on microbial and autoantigen antibody repertoire in adult-onset dermatomyositis patients sero-positive for TIF1 γ (TRIM33) autoantibodies. I have some comments and concerns as detailed below.

1) Power calculation: Did the authors perform any power calculation for the sample size and study design? How were the cases and controls selected and what were the inclusion and exclusion criteria?

2) Figure legends: Some of the main figure legends are just really long and wordy. Can they be succinct?

3) Statistical tests: For each of the analysis statistical tests must be explicitly mentioned whenever the p-values are mentioned.

The authors mention they tested for data normality but it is not clear which datasets were normal and which not. This also needs to be explained.

4) Figure 2: The original reads presented in Figure 2 was there any normalisation performed? In other words is there any bias in the sequence reads?

5) Pooled plasma: There is a concern on the use of pooled vs single plasma that leads to a compromise in quality and also biases in sequencing and over estimation in reads and so on.. I would suggest the authors to perform some level of validation to see how severe this impact and loss of sensitivity is.

6) The authors talk about, "DM pathogenesis depends on dynamic and personalised viral exposure..." It is a bit of a stretch and I would tone down the enthusiasm or provide more experimental evidence for the same. Words like "personalised" do have large implications which are not justified by data from this study.

7) We implemented the "Serum Antibody Repertoire Analysis (SARA)" pipeline (manuscript in preparation). Some more information on this method is necessary to evaluate its implications.

8) The authors use the term, "disease-specific" when they only show data for "disease-association". I would recommend to change this to disease-association or statistically significant association.

9) "Such a response is critical for host protection against pathogen expansion, and limits infection during the window of time needed to mount an effective specific (adaptive) immune response." I would avoid over-reaching statements like limits infection rather use reduces and in what effect size or fold change.

10) Multiple testing concerns: There is no apparent multiple testing performed. Why? there is a high chance of false positive associations given the high dimensional sequencing obtained

Note: Line numbering in reviewer responses refers to the corrected manuscript after incorporation of reviewers' comments.

Referee #1

Early onset dermatomyositis (DM) is an autoimmune disease that can be associated with auto-antibodies against TRIM33. In the current manuscript, Megremis et al utilize "Sara", a platform for identifying epitopes that are bound by the individual serum antibodies, in healthy controls and dermatomyositis patients, in order to identify epitopes (belonging either to microbes or self-proteins) that are specific or enriched in patients vs controls. The authors identify a large set of epitopes, many of which are enriched in one of the two groups. Particularly viral epitopes, TRIM proteins and type I interferon-induced proteins are found more often or uniquely in DM patients, and several epitopes share homology between TRIM proteins and poxviruses. The authors conclude that viral infections might be at the heart of the autoimmune disease.

The role of microbial infection in the aetiology of autoimmune disorders is intriguing and very much worth studying. The current manuscript provides a large number of possible leads that might help to unravel the cause and/or pathogenesis of DM, which is of great clinical importance and can be extrapolated to multiple other autoimmune diseases. I believe the used method is state-of-the-art and has the potential to identify, in an unbiased and high-throughput manner, relevant epitopes in various autoimmune diseases. However, I am somewhat worried about the conclusions that can be drawn from the data as presented by the authors, particularly whether the identification of an antigen that putatively binds a serum antibody actually means that the person actually was infected/exposed to the organism expressing the antigen, or that there might be a certain level of "noise". I have a number of questions/comments that may improve the current manuscript:

Response to reviewer 1: We thank reviewer 1 for investing time in going through this manuscript, especially during this challenging period. In this pilot study we have a first look at a quite unexplored area of systems immune research where the boundaries between the single pathogen causing disease and the microbial-viral exposome are not clear. We find the reviewers comments reasonable and constructive. We address them within the manuscript and in the following section:

1. The viral epitopes found suggest that the patients have been exposed to Variola virus (smallpox), which is eradicated but used in vaccines until 1967. Are there no patients or controls born after 1967?

Response to reviewer 1:

As can be seen in supplementary table 1, all the participants in this study had a date of birth before 1967, so at least for this study we do not have available plasma from individuals born after the global vaccination programs. Notably, immunity from smallpox vaccination lasts for decades, practically throughout the life expectancy (ref.15: Taub, D. *et al.* Immunity from smallpox vaccine persists for decades: a longitudinal study. *Am J Med* **121**, 1058-1064 (2008)). Moreover, it has been demonstrated that vaccinia antibodies are observed in non-immunised infants and that approximately 40% of mothers had significantly lower titers than their own newborn infants (Kempe, Yale J Biol Med. 1952 Feb; 24(4): 328-333: Passive Immunity to Vaccinia in Newborns I. Placental Transmission of Antibodies). Both

observations have also been verified in mouse models (Navarini A, et al. Long-lasting immunity by early infection of maternal-antibody-protected infants. *Eur J Immunol.* 2010;40(1):113-116). The above data suggest that there is a continuum of vaccinia/variola immune “signal” in individuals born after the eradication of smallpox. We aim to further investigate this in independent cohorts of DM and juvenile DM.

We have addressed the reviewer’s comment in the manuscript:

Line 333: “Notably, antibodies against vaccinia/variola virus are often observed in non-immunised infants of vaccinated mothers suggesting a continuum of VACV protection in the post-eradication era (added refs 16, 17).”

2. In addition to Variola, the DM patients are enriched in antibodies binding epitopes from a pretty aggressive set of viruses, including SARS, HIV, foot and mouth disease virus and HBV/HCV. Is this realistic? There must be a rich clinical history present when patients have such diseases. Do rare and highly pathogenic viruses more often infect future DM patients than healthy controls? Or is it possible that a number of the identified epitopes share accidental homology to these viruses?

Response to reviewer 1:

Infections of DM and polymyositis patients with highly pathogenic viruses have been reported in numerous cases as discussed in the discussion section (lines 335-343). Due to the role of TRIMs in innate immunity, this is also one of the factors that contributed to our hypothesis. Unfortunately, recording infection and/or vaccination history is not part of our standard DM clinical investigation. Integrating the available published data with our findings, we believe that due to the presence of TRIM and IFN-related autoantibodies TRIM33-autoantibody positive DM patients are prone to a higher level of viral exposure progressively expanding their antigen repertoire under the chronic burden of accumulated exposure. Whether this process is accompanied by a virus-related symptomatology (productive infection) is not yet known.

To this end, there is little available information regarding the role of the virome in asymptomatic and non-infectious individuals. The progress of shotgun metagenomics in the past decade, and hopefully in coming years, will provide new evidence regarding the role of asymptomatic presence of viral species in homeostasis and disease and is probably highly relevant to DM and other autoimmune disorders. In addition, the sequence-specific immune response at a global scale i.e. within the microbial-viral exposome, is still not well understood, so we do not know how exposure to one species might influence immunological reaction to another species, especially when these species are phylogenetically or taxonomically distant. Moreover, it has been demonstrated that even very short 5-mer peptides can often elicit strong antibody production, while the role of IgM and IgG in this process seems to be different, i.e. IgM tends to provide a more flexible system of sequence-specificity. Finally, sequence analysis is highly dependent on the reference databases used and their composition. For example, in the DM dataset we observed Bat coronavirus epitopes which when re-analysed with an updated reference database, were annotated as SARS-CoV-2 peptides (Megremis S, et al. Antibodies against immunogenic epitopes with high sequence identity to SARS-CoV-2 in patients with autoimmune dermatomyositis. *Ann Rheum Dis.* 2020;79(10):1383-1386). This is a good example of the complex sequence space.

Finally, the point of “accidental” homology as stated by the reviewer is exactly the problem in hand and falls within the remit of molecular mimicry. We are confident that in our study, “accidental” does not refer to the process of sequence classification since multiple steps have been taken to prevent accidental or random assignment of peptide sequences to different taxa. However, “accidental” can indeed refer to sequence homology between microbial and/or human proteins that can lead to molecular mimicry events important for human homeostasis and disease (Srinivasappa J, et al. Molecular mimicry: frequency of reactivity of monoclonal antiviral antibodies with normal tissues. *J Virol.* 1986;57(1):397-401).

3. The authors explain the presence of antibodies against Geminiviridae plant viruses through exposure to them via food. Why are they then enriched in the DM group? How is the presence of antigens from bacteriophages explained?

Response to reviewer 1:

Geminiviridae are plant viruses that are observed as part of both the respiratory and gut viromes, however they are more often detected in faecal samples (Rascovan N. Metagenomics and the Human Virome in Asymptomatic Individuals. *Annu Rev Microbiol.* 2016;70:125-141). We currently do not know what the role of these viruses is in DM or health or whether they reflect differences in feeding habits.

The presence of bacteriophage antigens is interesting. Phages themselves are indeed immunogenic microorganisms that can stimulate an adaptive immune response in humans (Krut O. Contribution of the Immune Response to Phage Therapy. *J Immunol.* 2018;200(9):3037-3044). Bacteriophages are active regulators of bacterial populations and changes in phage composition or diversity have been linked to various diseases including Inflammatory Bowel disease, Parkinson’s disease and Type 1 Diabetes (Sinha A. Bacteriophages: Uncharacterized and Dynamic Regulators of the Immune System. *Mediators Inflamm* 2019;2019:3730519). Even though the phage-bacteria prey-predator dynamics are not well understood, i.e. the target specificity of phages is constantly under co-evolution with their microbial hosts, it would be interesting to investigate the DM metagenome in various organs such as the respiratory system, the gut and the muscle. Moreover, the fact that we observed phage antigens being enriched mainly in the healthy group (Podoviridae and Myoviridae) could point towards a microbiome dysbiosis in DM that might be reflected by the increased presence of certain phages in HC that in turn contribute to the antibody maps as produced in our study. We have started looking into this aspect in combination with the bacterial antigens that we have identified but this is an ongoing study.

4. As the authors also state, the use of pooled plasma is a limitation of this study design. When pooling 10 or 20 sera from DM or HC, how do the authors exclude for the possibility that one patient skews the entire pool, creating the artificial appearance of ‘group-specific’ epitopes?

Response to reviewer 1:

Indeed, this is a limitation and we are currently designing a study to screen individual samples for a diverse repertoire of identified epitopes in order to measure the frequency of our observations within DM and HC. From a technical point of view, we have ensured that all plasma samples are equally represented since the same amount of isolated immunoglobulins was used during the pooling process and all subsequent steps were

normalised to the same level. Indirectly, if our enrichment was heavily skewed towards the profile of a single sample we believe that this would be evident in the microbial diversity observed when comparing the pools of 10 and 20 samples. As can be seen in figure S2C there is an expansion of the unique but not distinct microbial diversity in P20 compared to P10 in DM. This is also accompanied by an increase in the number of identified epitopes (Figure 2A, 2C, and 2E). This suggests that the differentiating signal between DM and HC is increasing with sample size. Given that the plasma samples used in P10 were also used in P20, then the sample under interrogation would have to be used only in P20, be extremely rich in epitopes from different species, and qualitatively distant from the healthy state. We cannot exclude this but it seems like an extreme scenario. Nevertheless, we aim to address this issue in our next study since we are very keen to follow up this work in larger cohorts of DM and juvenile-onset DM.

5. When looking for shared epitopes between viruses specific for DM and TRIM proteins, it might be informative to compare that to similar analyses done on the HC pool, or include other negative controls; this would eliminate the feeling of cherry picking.

Response to reviewer 1:

In the healthy group we identified only one Variola virus epitope in the HC pool of 10 samples. Increasing the pool to 20 samples resulted in the loss of enrichment of this epitope in the final dataset that was used to further investigate the virus antigens. Since our methodology is based on (1) competitive panning, (2) the healthy controls lack the specific TRIM autoantibodies, and (3) the pool of 20 healthy samples lack variola epitopes, we did not perform a similar sequence homology analysis. Moreover, we emphasize that we did not pick certain viruses for further investigation but we followed a data-driven approach (Figure 4D-4F). Specifically, Poxviruses were ranked in the top 5 viral species based on the normalised sequencing read counts per epitope and the richness of each viral family (Table S2 & S3).

6. Figure 1 and 2 are technical and provide a good overview of the results for the analyses. However, they could be combined or partially added to a supplemental figure.

Response to reviewer 1:

Due to the journal's limitation on the number of main and supplementary figures we cannot use these figures as supplementary information. Moreover, because of the manuscript's organisation with the methods section following results and discussion we feel that these figures are important for the reader to understand in a fast and efficient way the technical aspect of this study. Thus, if possible we would like to retain these figures in their current position.

7. Figures S2A and B look impressive but are quite useless at this resolution.

Response to reviewer 1:

We have uploaded Figures S2A and S2B as separate high-resolution files.

8. Line 170: "The over-representation of viral compared to cellular microbial species in the top-ranked microbes enriched in DM and HC suggests an important role of virus exposure in

the environment-host immune cross-talk in both healthy and disease states, and implies that viruses play an important role in the pathology of DM.” Why do the authors draw the last conclusion?

Response to reviewer 1:

Due to the competitive panning process, our final dataset of antigens in both DM and HC include variables that *a priori* maximize the distance between the two groups in a similar manner to differentially expressed genes, for example, in a typical transcriptome study. This means that enriched antigens observed in DM are in relation to HC and vice versa. The fact that we identify viral antigens enriched in the top 10% ranked species based on the $NGSR_{e_norm}$ in both DM and HC suggests that viral exposure has a significant role in DM (Figure 3). Additional potential differentiating cellular microbe antigens will be investigated in the future.

We have addressed the reviewer’s comment in the manuscript:

Line 167: “Due to the competitive panning process, the over-representation of viral compared to cellular microbial species in the top-ranked microbes enriched in DM and HC suggests an important role of virus exposure in the environment-host immune cross-talk in DM.”

Line 322: “The competitive design of the epitope enrichment process suggests that viral exposure has a significant role in DM; additional potential differentiating cellular microbes will be further investigated in the future.”

Reviewer #2 (Remarks to the Author):

In the manuscript “Microbial and autoantibody immunogenic repertoires in TIF1 γ autoantibody positive dermatomyositis” Spyridon Megremis, et al, report the anti-microbial and anti-autoantigen antibody repertoire in patients with adult-onset dermatomyositis that are sero-positive for TIF1 γ (TRIM33) autoantibodies. The authors use a very interesting high throughput approach to identify the IgOme against microbial and human epitopes, with an increased diversity of antibodies directed to microbial epitopes, particularly viruses of the Poxviridae family. Further, the authors indicate that the autoantibodies identified are directed against a large portion of the human proteome, and emphasize on TRIM proteins which have homology with viruses, including poxviruses. Thus, they claim molecular mimicry and epitope spreading as having a potential significant role in the pathogenesis of dermatomyositis. However, I have a few concerns that need to be addressed to strengthen this report.

We would like to thank reviewer 2 for their thorough and constructive comments. We address them within the manuscript and in the following section:

Title.

The manuscript reports the repertoire of epitopes recognized for antibodies in pooled plasma of dermatomyositis patients as compared with pooled plasma from healthy controls. With this in mind, to title the manuscript as “the microbial and autoantibody immunogenic repertoire”, seems somehow confusing. It needs to be clear that what is reported is the repertoire of antibodies recognizing these antigens.

This holds true for the entire manuscript in which most of the time the text reads as if microbiome techniques were used.

Response to reviewer 2: We have changed the terminology throughout the paper (indicated by tracked changes) and the title: “Analysis of total antibody repertoires in TIF1 γ autoantibody positive dermatomyositis”.

Results.

1. Subheading: Higher microbial diversity in dermatomyositis. This report does not report on microbial diversity, but on the antibody repertoire. Thus, the data does not demonstrate the presence of a stably-enriched microbial component in DM, especially when one of the main claims is the potential molecular mimicry and epitope spreading.

Response to reviewer 2: We have changed the subheading to: “Wider repertoire of antibodies recognizing microbial antigens in dermatomyositis” and tracked changes relating to this comment within the text. We have also removed the statement regarding the stably-enriched microbial component in DM.

2. It is important to state the rationale for the use of two different pools in each group.

Response to reviewer 2: Line 96: “We used two pools of increasing sample sizes so that we could evaluate the heterogeneity of our observations within the DM and HC groups”.

3. Poxiviruses are tightly linked with dermatomyositis. Antibodies directed against dsDNA viruses, group in which poxiviruses are included, are contained in a higher proportion in the HC pools. However, when each pool was individually evaluated for antibodies against each family, Poxviridae was the only family that ranked highly in the DM pool. Amongst these the highest was variola virus. As mentioned later in the manuscript, the likelihood of these patients having been vaccinated is high, however, it is important to include the actual vaccination status of these patients and controls (as best possible) in the corresponding table.

Response to reviewer 2: As we have commented in the discussion section, and in response to reviewer 1, unfortunately vaccination or infection history was not available for the samples included in this study.

4. While I assume the data will be made publicly available, it is important to include the unsupervised identification of human protein targets for reference (maybe as a supplementary figure). Where those TRIM and IFN-regulated proteins rank? Are they in among the most significant hits? Is there a differential signature between the two pools?

Response to reviewer 2: We have included figures S5E & S5F depicting the distribution of all identified autoantibody protein targets in DM P20 and HC P20. This shows the distribution of annotation bitscores vs mean NGS enrichment (blue nodes). The annotation bitscores were computed per each mapped epitope by a stats framework that ranks protein matches based upon epitope-protein annotation confidence; intra-protein discontinuous sequence matches; redundancy of protein isoforms and common sequence matches; single-epitope multi-protein parsing, and set analysis of resultant patient human protein lists. Annotated human protein rankings were then tuned as a function of each epitope mean NGS read count. This count was generated from the read depth of each noise-floor-removed divergent NGS sequence that translated to the same specific amino acid sequence. Finally set-analysis was used to inspect

the degree of overlap of any entities common to DM and HC patients. We have annotated TRIM proteins with red colour and IFNG-related proteins with yellow.

As we have stated in Line 223 there is a differential signature between DM and HC: “From our autoantibody analysis, in DM P20 we identified antibodies against a total of 2885 human protein targets, of which 2537 were highly specific sequence annotation hits (Mean Sig.< 0.578). In HC P20 we detected 1450 human protein targets, 811 with high specificity (Mean Sig.< 0.55). Autoantibodies identified in both DM and HC constituted only 13% (n=498) of total proteins and 8.4% (n=260) of highly specific proteins suggesting a strong disease-specific proteome signature. The 2:1 (total proteins) and 3:1 (high specificity) ratios of identified autoantibodies in DM over HC, suggest that an expanded subset of the human proteome is targeted by autoantibodies in DM patients compared to HC.” Moreover, since the acquired data originate from competitive biopanning they a priori represent the differential signature between the two clinical groups.

5. The authors identify a series of epitopes that are shared between poxiviruses and TRIM3 in DM. I think that demonstration of the capability of these plasma samples to recognize TRIM3 (IP, WB, ELISA?) would shed insight on the relevance of these antibodies in the etipopathogenesis. Also, as indicated in the discussion (line 382), TRIM3 is considered as an inhibitor of liver, colorectal and gastric cancer; however, only one of the patients included in the study have DM-associated liver cancer associated. What are the potential implications of this seemingly discrepancy? Please discuss.

Response to reviewer 2: We agree with the reviewer and we will evaluate TRIM3 along with a selection of human and microbial epitopes in independent cohorts of IIMs and DM as a follow up to this study. In the discussion, we note that TRIM proteins with both oncogenic and tumour suppressor roles often show dysregulated activity that has been associated with the development of human cancer. Based on the relatively small number of DM patients included in this study and the different malignancies reported in cancer-associated myositis, we do not believe that strong conclusions can be drawn based on the observation of hepatic cancer in 1/9 cancer-associated myositis patients. We have modified the discussion (line 387): “Identification of autoantibodies against cancer-associated TRIMs accords with the strong temporal association between myositis and development of malignancies in adult-onset anti-TIF1 γ positive DM⁴, consistent with the high proportion of cancer-associated myositis cases in the current study (45%), including ovarian, lung and hepatic cancer.”

6. Pooling plasma from patients with DM-associated cancer and those cancer-free, may be skewing the results toward a different autoantibodies repertoire, thus I believe comparison of these two groups is crucial.

Response to reviewer 2: We acknowledge this limitation of our study using pooled plasma. As noted in our response to reviewer 1, we are currently designing a study to screen a larger number of individual samples for a diverse repertoire of identified epitopes in order to measure the frequency of our observations within DM and HC; this study will also enable us to compare the two groups of cancer-associated DM and DM without cancer.

7. The authors suggest the accumulation of antibodies against proteins that strongly contribute to IFN gamma signaling and the broader antiviral mechanism by IFN-regulated proteins in DM patients (lines 243-245). Do these patients have more frequent viral

infections? How can this be aligned with what seems to be a potential ability to mount better antiviral responses?

Response to reviewer 2: We have addressed a similar question from reviewer 1. Please see response to reviewer 1 point 2.

Specifically for reviewer 2 comments: We are not sure whether the accumulation of antibodies that target IFN gamma signalling of IFN-regulated proteins qualifies as a “better” antiviral response. We believe that the non-random accumulation of autoantibodies against IFN-regulated proteins in DM can jeopardise an optimal innate and adaptive immune response. This was also recently demonstrated in COVID-19 disease where autoantibodies against type I IFNs were observed in patients with life-threatening COVID-19 (*Science* 24 Sep 2020: eabd4585). Moreover, the presence of antibodies against a wider repertoire of microbial species can obviously reflect a good immune response but the fact that we identify a higher number of targeted microbial species might also reflect higher exposure at least for these species. We could possibly draw a more conclusive view if indeed the number of reported infections was monitored but unfortunately infection history is usually not part of the DM epidemiological data, something that we address in the discussion section.

Discussion

1. As mentioned before, it is important to make it clear that the data shown is on the antibodies against microbial epitopes and human proteins; no microbiome studies were performed. For instance, Viruses were not over-represented (Line 321), but antibodies recognizing viruses were over-represented; Enriched families in DM included... (line 323); DM plasma also was enriched in viruses reported to affect... (line 335); etc. Please be sure that this is clarified in the entire manuscript.

Response to reviewer 2: We have addressed this throughout the manuscript, as shown by tracked changes.

2. The authors acknowledge the limitation of using pooled plasma. In this regard, it is very important to differentiate between patients with DM-associated cancer, which they would not be able to accomplish with the strategy undertaken.

Response to reviewer 2: Please see previous response to comment 6.

Methods

1. There are no epitopes in pooled sera, but antibodies against those epitopes (Line 479)
2. There were no pooled sera but plasma (line 483).

Response to reviewer 2: please see changes addressed in manuscript

Figures

1. Figure 1. This figure indicates work with pooled sera, but all the text indicates pooled plasma; which was used?
2. Figure 3 (legend). Ranking of identified microbial species... This manuscript does not identify microbial species, but the epitopes to which plasma antibodies are directed against.

Response to reviewer 2: We have used plasma and not serum. The confusion is due to the name of the SARA pipeline. We have corrected it in the manuscript. Figure 3 legend is also

corrected.

Supplementary table 1

1. The column for ethnicity contains mixed information, i.e. ethnicity (White British), and race (Caucasian). This needs to be corrected, or eliminated the authors need to consider that “oriental” is neither a race nor an ethnicity, thus this term needs to be replaced for the correct one. However, the authors need to explain what the importance of this information is, since all the patients and controls, but one of the patients seem to be of the same race? In fact, one wonders whether the inclusion of only one patient from a different racial group is adequate, or represents a further limitation of the study in terms of both lack of diversity or the potential of skewing the IgOme in a way that is not representative of the racial majority included; please discuss.

Response to reviewer 2: We thank the reviewer for identifying this error. This information has been corrected in supplementary table 1. The patients and controls were selected to be racially matched, given the small sample size in this initial study. Inclusion of one non-Caucasian patient was due to a sample selection error; However we do not feel that inclusion of this one patient will significantly skew the results, and we have included this information on race for transparency.

2. Please clarify the cancer associated myositis definition in the table legend. Is this defined as cancer that was diagnosed within three years after the onset of myositis?

Response to reviewer 2: The table legend has been modified to say “Cancer-associated myositis defined as cancer diagnosed within three years (before or after) of myositis onset.”

3. Considering the importance of TRIM proteins as oncogenic proteins (TRIM27, TRIM47) or tumor suppressors (TRIM3), and the fact that the healthy control pools did not have antibodies against epitopes in these proteins, it would be very informative to establish the malignancy status in the set of healthy subjects.

Response to reviewer 2: Healthy controls were free of malignancy at the time of sample collection.

Supplementary tables 2 and 3

1. Please clarify that is not richness of viral families but the antibodies against them.

Response to reviewer 2: Corrected in manuscript

Reviewer #3 (Remarks to the Author):

I have read with interest the manuscript titled, "Microbial and autoantibody immunogenic repertoires in TIF1 γ autoantibody positive dermatomyositis". The paper focuses on microbial and autoantigen antibody repertoire in adult-onset dermatomyositis patients sero-positive for TIF1 γ (TRIM33) autoantibodies. I have some comments and concerns as detailed below.

We would like to thank reviewer 3 for the reasonable and constructive comments. We address them within the manuscript and in the following section:

1) Power calculation: Did the authors perform any power calculation for the sample size and study design? How were the cases and controls selected and what were the inclusion and exclusion criteria?

Response to reviewer 3: Sample size was pragmatic based on available TIF1 positive patient numbers, and previous proof of concept studies of the SARA approach carried out in ovarian cancer and childhood acute lymphoblastic leukaemia patients which used pooled patient sera from 8 and 100 patients respectively. These proof of concept studies were carried out to determine the most informative pool size based on intra- and inter-pool variance, balancing heterogeneity in smaller pools versus masking of notable infections in larger sample pools. Salient predictions derived from these pooled-sera were subsequently assessed by molecular assays conducted upon those individual patient samples which were originally pooled (and additional patient samples). (manuscript in preparation).

Since this area of research is relatively unexplored it is hard to envision appropriate pooled sample sizes for different disease groups. This is because the accumulation of antibodies probably varies between groups of individuals with different pathophysiological, epidemiological and socioeconomic characteristics. Larger studies with the aim to model the accumulation of antibodies with increasing sample size in different populations are necessary. Nevertheless, our data suggest that increasing the sample size from 10 to 20 samples seems to cover this healthy group where the number and type of microbial species targeted by the antibodies remains relatively stable (phylogenetic tree, Fig. S2B), whereas in the DM pools, antibodies against a wider microbial repertoire was observed (phylogenetic tree, Fig S2A). Based on this we believe that the use of the P20 samples are indeed more informative than the P10s and more appropriate to draw conclusions in DM.

As described in the methods, plasma samples were collected from anti-TIF1 positive adult-onset dermatomyositis (DM) patients through the UK Myositis Network. All individuals fulfilled definite or probable Bohan and Peter classification criteria for dermatomyositis. The following sentence has been added “Samples were selected at random and based on plasma availability.” “Gender and age matched (at time of sample collection) healthy controls (HC) were randomly selected from the University of Manchester Longitudinal Study of Cognition in Normal Healthy Old Age cohort.”

2) Figure legends: Some of the main figure legends are just really long and wordy. Can they be succinct?

Response to reviewer 3: We have reduced figure legends for figure 1 and figure S1, the ones with the largest figure legends.

3) Statistical tests: For each of the analysis statistical tests must be explicitly mentioned whenever the p-values are mentioned.

Response to reviewer 3: We have added specific description of the statistical tests used whenever a statistical significance is mentioned within the manuscript.

The authors mention they tested for data normality but it is not clear which datasets were normal and which not. This also needs to be explained.

Response to reviewer 3: We have described in the methods section (lines 575-584) the statistical tests used based on the normality distribution of the data. Thus, within the manuscript we state the statistical test. For example, when a Two tailed t test is mentioned this refers to data normally distributed versus Mann Whitney used for data not normally distributed.

4) Figure 2: The original reads presented in Figure 2 was there any normalisation performed? In other words is there any bias in the sequence reads?

Response to reviewer 3:

The original NGS reads that mapped to the AA epitopes represent NGS DNA sequences of biopanned-enriched plasmid variance regions that are expressed as antigen epitopes. In this regard our NGS sequencing is not concordant with typical direct sequencing of infectious agents (for example metagenomics of soil samples) since the purpose of biopanning is to preferentially amplify immunogenically relevant antigen sequences and deplete immunogenically irrelevant sequences prior to NGS loading.

Figure S3E denotes tightly-defined OD readings of FliTrx *E coli* during biopanning wherein no differences were observed between DM and HC cross-panning pairs (Welch's t test p values of 0.861 (P20 cohorts) and 0.954 (P10 cohorts)). Plasmid yields were normalised and QC'd across each cohort prior to library preparation. Library preparations were then QC'd through standard methods.

Sequence bias that can occur in routine NGS (such as gene length bias or genome mapability bias) are not typically relevant to our approach since we sequence short and identical length plasmid sequences. Technical read bias is theoretically possible from low complexity reads hindering accurate base calling across the flow cell events, and we control for this by incorporating barcoded plant complexity genomes which are removed during FASTQ file deconvolution.

Any potential sequencing bias derived from inter-user biopanning or different sequence runs was previously empirically determined ("Serum Antibody Repertoire Analysis (SARA)" pipeline (manuscript in preparation)) and presents less than 5% differences at the epitope resolution - predominantly at non-informative low enrichment - and none of which confers any change in significance or qualitative difference or on downstream analysis. For Distinct NGS reads (Figure 2D) any sequence data 3' of stop codons was removed and read counts recalculated since mechanistically these denote identical antigen biopan captures. Phred scores were recalculated and expressed region sequences containing low quality base calls were removed. Low read count of resultant distinct sequences were subsequently filtered by our described $10^{-\sigma-99}$ threshold to remove biopan-derived noise floor (i.e. low read count NGS reads). This is required as a consequence of residual buffer run off inherent to all bench-lab biopanning assays since it contains immunogenically-irrelevant random reads.

5) Pooled plasma: There is a concern on the use of pooled vs single plasma that leads to a compromise in quality and also biases in sequencing and over estimation in reads and so on.. I would suggest the authors to perform some level of validation to see how severe this impact and loss of sensitivity is.

Response to reviewer 3: We agree with the reviewer and we will evaluate TRIM3 along with a selection of human and microbial epitopes in independent cohorts of IIMs and DM as a follow up to this study. We have acknowledged the limitation of pooling samples in the discussion section.

6) The authors talk about, "DM pathogenesis depends on dynamic and personalised viral exposure..." It is a bit of a stretch and I would tone down the enthusiasm or provide more experimental evidence for the same. Words like "personalised" do have large implications which are not justified by data from this study.

Response to reviewer 3: we have changed this into "We propose a model whereby DM pathogenesis depends on viral exposure patterns, which start early in life". (Line 414)

7) We implemented the "Serum Antibody Repertoire Analysis (SARA)" pipeline (manuscript in preparation). Some more information on this method is necessary to evaluate its implications.

Response to reviewer 3: The manuscript is currently under preparation. We feel that we have provided adequate and in-depth information about this analytical pipeline describing step-by-step the different stages of the analysis and providing the appropriate metrics for our study.

8) The authors use the term, "disease-specific" when they only show data for "disease-association". I would recommend to change this to disease-association or statistically significant association.

Response to reviewer 3: We have replaced disease-specific with disease-associated: Lines 83, 227, 449, Figure 4 legend, Figure 7 legend, Figure S6 legend,

9) "Such a response is critical for host protection against pathogen expansion, and limits infection during the window of time needed to mount an effective specific (adaptive) immune response."

I would avoid over-reaching statements like limits infection rather use reduces and in what effect size or fold change.

Response to reviewer 3: We have corrected this in the manuscript Line 70.

10) Multiple testing concerns: There is no apparent multiple testing performed. Why? there is a high chance of false positive associations given the high dimensional sequencing obtained

Multiple testing was performed for all comparisons between three or more groups. This is stated in the methods section and in the figure legends that show statistical differences. Specifically, the Kruskal-Wallis nonparametric test was used to compare three or more groups and the p value reported is adjusted based on the Dunn's multiple comparisons test to correct for multiple comparisons.

REVIEWERS' COMMENTS:

Reviewer #3:

None

Reviewer #4:

Remarks to the Author:

I truly enjoyed this revision of the manuscript "Analysis of total antibody repertoires in TIF1y autoantibody positive dermatomyositis" by Megremis et al. AS mentioned before this is a very interesting report on the IgOme against microbial and human epitopes utilizing a novel approach. The results reported in this manuscript clearly demonstrated that, there is an increased diversity of antibodies directed to microbial epitopes, particularly of the Poxviridae family in two pooled plasma samples of DM patients. Additionally, the authors identified autoantibodies directed against a large portion of the human proteome, especially TRIM proteins, which show homology with different viruses, suggesting molecular mimicry and epitope spreading as a potential major player in the pathogenesis of DM. I believe, there is much potential in this methodology to continue not only the studies in DM, but also in other antibody mediated autoimmune diseases. In this revised version, the authors have addressed and discussed my previous comments and concerns. I commend the authors for the great effort they have put into this work. However, I do still have a couple of minor comments that I hope will be helpful.

1. Although the potential confusion in serum/plasma usage has been clarified and corrected where appropriate, on the top of the right column of figure 1A it still says sera pool. I do understand that this is the description of the original methodology which as its name indicates is based on serum antibody repertoire analysis (SARA), however, I think it will be better to change that to plasma to adapt to the present report.
2. I appreciate the addition of panels E and F in supplementary figure 5, it really helps understanding the analysis of protein targets. I recommend to organize the figure so the panels flow from A to F without disruption. This would make it easy for the readers.
3. The authors have corrected supplementary table 1 indicating that all the patients but one, are Caucasian. In my previous comment I asked about the possibility of this unique patient skewing the data, and the authors have reasonably clarified why they do not think this would be the case. However, as they mentioned, for the sake of clarity, it would be good to include a statement in the discussion indicating this fact.

Reviewer #5:

Remarks to the Author:

The authors have satisfactorily addressed my comments